# Structure in the variability of the basic reproductive number ($R_0$) for Zika epidemics in the Pacific islands

Clara Champagne[1,2]*, David Georges Salthouse[1], Richard Paul[3,4], Van-Mai Cao-Lormeau[5], Benjamin Roche[6], Bernard Cazelles[1,6]*

[1]IBENS, UMR 8197 CNRS-ENS Ecole Normale Supérieure, Paris, France; [2]CREST, ENSAE, Université Paris Saclay, France; [3]Department of Genomes and Genetics, Institut Pasteur, Unité de Génétique Fonctionnelle des Maladies Infectieuses, Paris, France; [4]Centre National de la Recherche Scientifique, URA 3012, Paris, France; [5]Unit of Emerging Infectious Diseases, Institut Louis Malardé, Tahiti, France; [6]International Center for Mathematical and Computational Modeling of Complex Systems (UMMISCO), UPMC/IRD, Bondy cedex, France

**Abstract** Before the outbreak that reached the Americas in 2015, Zika virus (ZIKV) circulated in Asia and the Pacific: these past epidemics can be highly informative on the key parameters driving virus transmission, such as the basic reproduction number ($R_0$). We compare two compartmental models with different mosquito representations, using surveillance and seroprevalence data for several ZIKV outbreaks in Pacific islands (Yap, Micronesia 2007, Tahiti and Moorea, French Polynesia 2013-2014, New Caledonia 2014). Models are estimated in a stochastic framework with recent Bayesian techniques. $R_0$ for the Pacific ZIKV epidemics is estimated between 1.5 and 4.1, the smallest islands displaying higher and more variable values. This relatively low range of $R_0$ suggests that intervention strategies developed for other flaviviruses should enable as, if not more effective control of ZIKV. Our study also highlights the importance of seroprevalence data for precise quantitative analysis of pathogen propagation, to design prevention and control strategies.

*For correspondence:
champagn@biologie.ens.fr (CC);
cazelles@biologie.ens.fr (BC)

**Competing interests:** The authors declare that no competing interests exist.

## Introduction

In May 2015, the first local cases of Zika were recorded in Brazil and by December of the same year the number of cases had surpassed 1.5 million. On February 2016, the World Health Organization declared Zika as a public health emergency of international concern (*Who, 2016*) and in March 2016, local transmission of Zika was recognized in 34 countries. Previously the Zika virus had circulated in Africa and Asia but only sporadic human cases had been reported. In 2007 the outbreak on Yap (Micronesia) was the first Zika outbreak outside Africa and Asia (*Duffy et al., 2009*). Since, Zika outbreaks have been also reported in French Polynesia and in New Caledonia (*Cao-Lormeau et al., 2014*; *Dupont-Rouzeyrol et al., 2015*) between 2013 and 2014 and subsequently, there have been cases of Zika disease in the Cook Islands, the Solomon Islands, Samoa, Vanuatu, and Easter Island (Chile) (see Figure 1 in *Petersen et al. [2016]*).

Zika virus (ZIKV) is a flavivirus, mostly transmitted via the bites of infected Aedes mosquitoes, although non-vector-borne transmission has been documented (sexual and maternofoetal transmission, laboratory contamination, transmission through transfusion) (*Musso and Gubler, 2016*). The most common clinical manifestations include rash, fever, arthralgia, and conjunctivitis (*Musso and Gubler, 2016*) but a large proportion of infections are asymptomatic or trigger mild symptoms that can remain unnoticed. Nevertheless, the virus may be involved in many severe

**eLife digest** Zika virus is an infectious disease primarily transmitted between people by mosquitoes. While most people develop mild flu-like symptoms, infection during pregnancy can interfere with how the baby's head and brain develop. Until recently, the virus had only been seen sporadically in Africa and Asia, but since 2007, outbreaks have been recorded on several Pacific islands. In 2015, the Zika virus reached the Americas, and within six months over 1.5 million cases had been reported in Brazil alone.

There is an urgent need to understand how the Zika virus moves within a population in order to help policymakers, and public health professionals, plan treatment and control of outbreaks of the disease. Researchers often use predictive models to estimate how a disease will spread. A parameter commonly calculated by these models is the "basic reproductive number", or $R_0$, which represents the average number of additional cases of the disease caused by one infected individual.

Using models that incorporated data from Zika virus outbreaks that occurred on several Pacific islands, Champagne et al. have produced estimates of $R_0$ that range from 1.5-4.1.

The $R_0$ values are greater than one, indicating that infection will spread within a population, but in the same range as those obtained for dengue fever, another closely related mosquito-borne disease. This suggests that by taking appropriate measures, the spread of Zika and dengue can be controlled to similar extents.

A closer look at the relationship between the population size and the predicted $R_0$ value for each Pacific island revealed an unexpected inverse relationship: the smaller the population, the larger the value of $R_0$. Since other regional factors may also explain these large differences between settings, further work is needed to disentangle context-specific from disease-specific factors. In this respect, data about seroprevalence (the number of people whose blood shows evidence of a past infection) in different populations is crucial for precisely analyzing the spread of Zika virus.

neurological complications, including Guillain-Barre syndrome (*Cao-Lormeau et al., 2016*) and microcephaly in newborns (*Schuler-Faccini et al., 2015*). These complications and the impressive speed of its geographically propagation make the Zika pandemic a public health threat (*Who, 2016*). This reinforces the urgent need to characterize the different facets of virus transmission and to evaluate its dispersal capacity. We address this here by estimating the key parameters of ZIKV transmission, including the basic reproduction number ($R_0$), based on previous epidemics in the Pacific islands.

Defined as the average number of secondary cases caused by one typical infected individual in an entirely susceptible population, the basic reproduction number ($R_0$) is a central parameter in epidemiology used to quantify the magnitude of ongoing outbreaks and it provides insight when designing control interventions (*Diekmann et al., 2010*). It is nevertheless complex to estimate (*Diekmann et al., 2010*; *van den Driessche and Watmough, 2002*), and therefore, care must be taken when extrapolating the results obtained in a specific setting, using a specific mathematical model. In the present study, we explore the variability of $R_0$ using two models in several settings that had Zika epidemics in different years and that vary in population size (Yap, Micronesia 2007, Tahiti and Moorea, French Polynesia 2013–2014, and New Caledonia 2014). These three countries were successively affected by the virus, resulting in the first significant human outbreaks and they differ in several ways, including population size and location specific features. Hence, the comparison of their parameter estimates can be highly informative on the intrinsic variability of $R_0$. For each setting, we compare two compartmental models using different parameters defining the mosquito populations. Both models are considered in a stochastic framework, a necessary layer of complexity given the small population size and recent Bayesian inference techniques (*Andrieu et al., 2010*) are used for parameter estimation.

## Results

We use mathematical transmission models and data from surveillance systems and seroprevalence surveys for several ZIKV outbreaks in Pacific islands (Yap, Micronesia 2007 (*Duffy et al., 2009*), Tahiti

and Moorea, French Polynesia 2013–2014 (*CHSP, 2014*; *Mallet et al., 2015*; *Aubry et al., 2015b*), New Caledonia 2014 [*DASS, 2014*]) to quantify the ZIKV transmission variability.

Two compartmental models with vector-borne transmission are compared (cf. *Figure 1*). Both models use a Susceptible-Exposed-Infected-Resistant (SEIR) framework to describe the virus transmission in the human population, but differ in their representation of the mosquito population. *Figure 1a* is a schematic representation derived from *Pandey et al. (2013)* and formulates explicitly the mosquito population, with a Susceptible-Exposed-Infected (SEI) dynamic to account for the extrinsic incubation period (time taken for viral dissemination within the mosquito).

By contrast, in the second model (*Figure 1b*) based on *Laneri et al. (2010)* the vector is modeled implicitly: the two compartments $\kappa$ and $\lambda$ do not represent the mosquito population but the force of infection for vector to human transmission. This force of infection passes through two successive stages in order to include the delay associated with the extrinsic incubation period: $\kappa$ stands for this latent phase of the force of infection whereas $\lambda$ corresponds directly to the rate at which susceptible humans become infected.

The basic reproduction number of the models ($R_0$) is calculated using the next Generation Matrix method (*Diekmann et al., 2010*):

$$R_0^{Pandey} = \sqrt{\frac{\beta_H \beta_V \tau}{\gamma \mu (\mu + \tau)}}$$

$$R_0^{Laneri} = \sqrt{\frac{\beta}{\gamma}}$$

In addition, we consider that only a fraction $\rho$ of the total population is involved in the epidemic, due to spatial heterogeneity, immuno-resistance, or cross-immunity. For both models we define $N = \rho \cdot H$ with H the total size of the population reported by census.

The dynamics of ZIKV transmission in these islands is highly influenced by several sources of uncertainties. In particular, the small population size (less than 7000 inhabitants in Yap) leads to high variability in transmission rates. Therefore all these models are simulated in a discrete stochastic framework (Poisson with stochastic rates [*Bretó et al., 2009*]), to take this phenomenon into account. Stochasticity requires specific inference techniques: thus estimations are performed with PMCMC algorithm (particle Markov Chain Monte Carlo [*Andrieu et al., 2010*]).

Using declared Zika cases from different settings, the two stochastic models (*Figure 1*) were fitted (*Figures 2–3*). These models allow us to describe the course of the observed number of cases and estimate the number of secondary cases generated, $R_0$. Our estimates of $R_0$ lie between 1.6 (1.5–1.7) and 3.2 (2.4–4.1) and vary notably with respect to settings and models (*Figures 2–3* and *Tables 1–2*). Strikingly, Yap displays consistently higher values of $R_0$ in both models and in general, there is an inverse relationship between island size and both the value and variability of $R_0$. This phenomenon may be explained by the higher stochasticity and extinction probability associated with smaller populations and can also reflect the information contained in the available data. However, the two highly connected islands in French Polynesia, Tahiti and Moorea, display similar values despite their differing sizes.

Regarding model variability, $R_0$ estimates are always higher and coarser with the Pandey model than with the Laneri model (cf. *Tables 1–2*). The Pandey model has two additional estimated parameters (in particular, the mosquito lifespan), which can explain the higher variability of the output. It is worth noting that these parameters are very sensitive (see Materials and methods). The difference in $R_0$ may also be linked to the difference in the estimated initial number of infected individuals ($H_I(0)$), which is higher in the Laneri model than in the Pandey model. Because of the high proportion of asymptomatic cases (the ratio of asymptomatic:symptomatic is estimated to be 1:1.3, V.-M Cao-Lormeau personal communication), it is hard to determine which scenario is more realistic, the time between introduction of the disease into the island and the first reported symptomatic case being unknown in most settings.

For the durations of infectious and intrinsic incubation (in human) and extrinsic incubation (in mosquito) periods, the posterior density ressembles the informative prior (cf. Figures 6–13), indicating the models' incapacity to identify properly these parameters without more informative data.

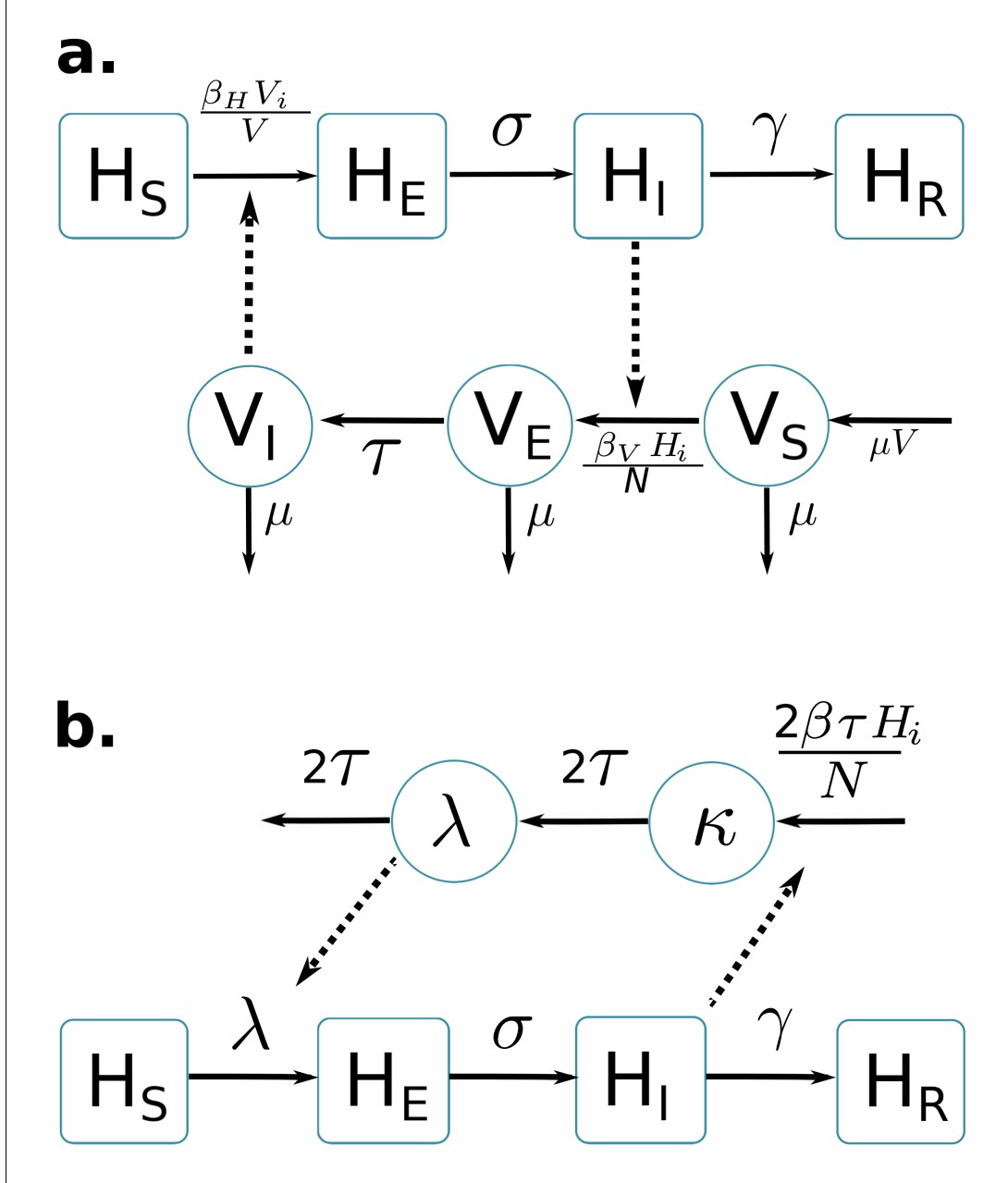

**Figure 1.** Graphical representation of compartmental models. Squared boxes and circles correspond respectively to human and vector compartments. Plain arrows represent transitions from one state to the next. Dashed arrows indicate interactions between humans and vectors. (a) Pandey model (*Pandey et al., 2013*). $H_S$ susceptible individuals; $H_E$ infected (not yet infectious) individuals; $H_I$ infectious individuals; $H_R$ recovered individuals; $\sigma$ is the rate at which $H_E$-individuals move to infectious class $H_I$; infectious individuals ($H_I$) then recover at rate $\gamma$; $V_S$ susceptible vectors; $V_E$ infected (not yet infectious) vectors; $V_I$ infectious vectors; $V$ constant size of total mosquito population; $\tau$ is the rate at which $V_E$-vectors move to infectious class $V_I$; vectors die at rate $\mu$. (b) Laneri model (*Laneri et al., 2010*). $H_S$ susceptible individuals; $H_E$ infected (not yet infectious) individuals; $H_I$ infectious individuals; $H_R$ recovered individuals; $\sigma$ is the rate at which $H_E$-individuals move to infectious class $H_I$; infectious individuals ($H_I$) then recover at rate $\gamma$; implicit vector-borne transmission is modeled with the compartments $\kappa$ and $\lambda$; $\lambda$ current force of infection; $\kappa$ latent force of infection reflecting the exposed state for mosquitoes during the extrinsic incubation period; $\tau$ is the transition rate associated to the extrinsic incubation period.

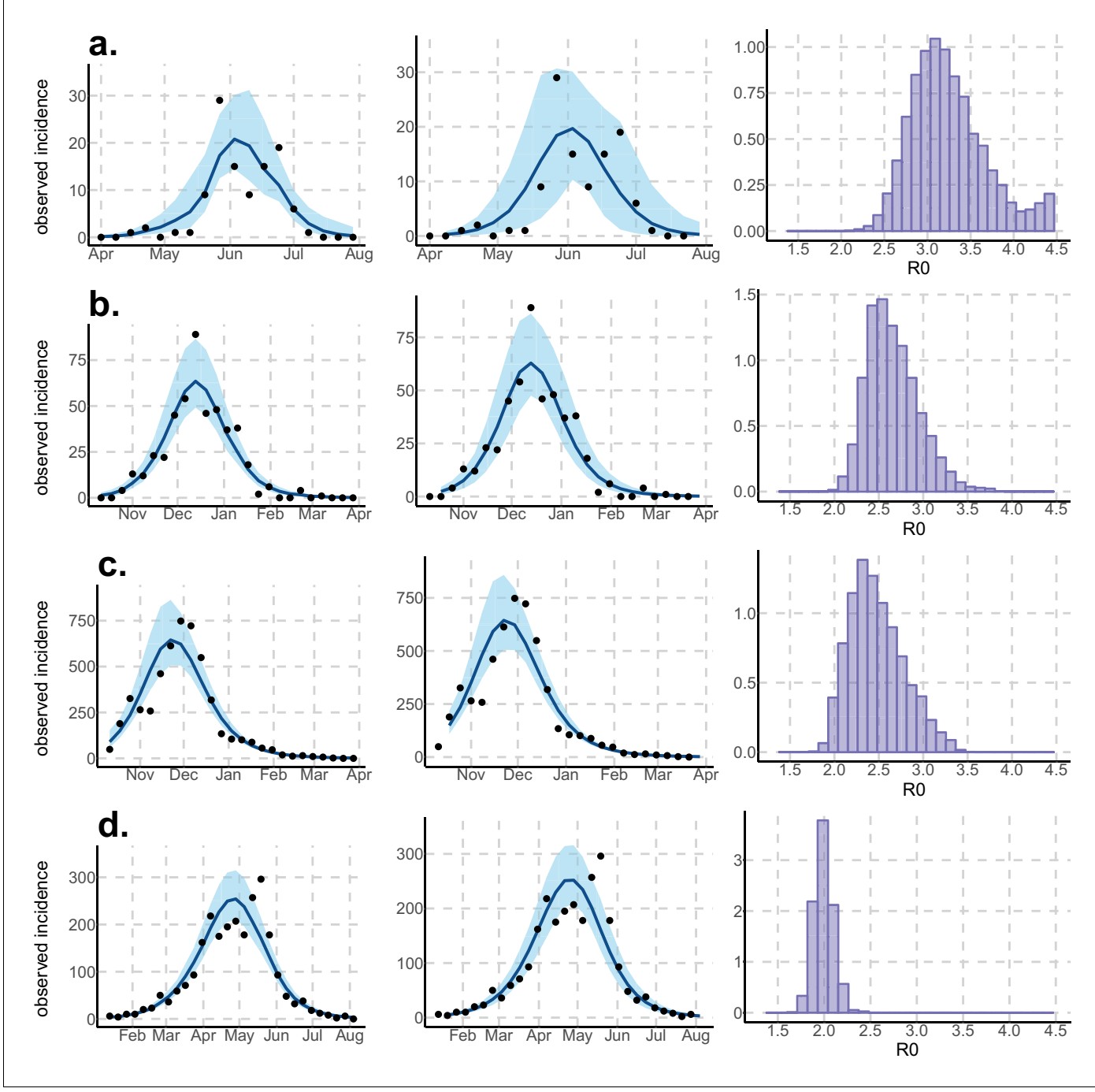

**Figure 2.** Results using the Pandey model. Posterior median number of observed Zika cases (solid line), 95% credible intervals (shaded blue area) and data points (black dots). First column: particle filter fit. Second column: Simulations from the posterior density. Third column: $R_0$ posterior distribution. (a) Yap. (b) Moorea. (c) Tahiti. (d) New Caledonia. The estimated seroprevalences at the end of the epidemic (with 95% credibility intervals) are: (a) 73% (CI95: 68–77, observed 73%); (b) 49% (CI95: 45–53, observed 49%); (c) 49% (CI95: 45–53, observed 49%); (d) 39% (CI95: 8–92). See *Figure 4*.

Moreover, these parameters have a clear sensitivity (see Materials and methods) and precise field measures are therefore crucial for reliable model predictions.

The fraction $\rho$ of the population involved in the epidemic is well estimated when the seroprevalence is known (in Yap and French Polynesia). In these cases, the proportion of the population involved is slightly greater than the seroprevalence rate, indicating a very high infection rate among

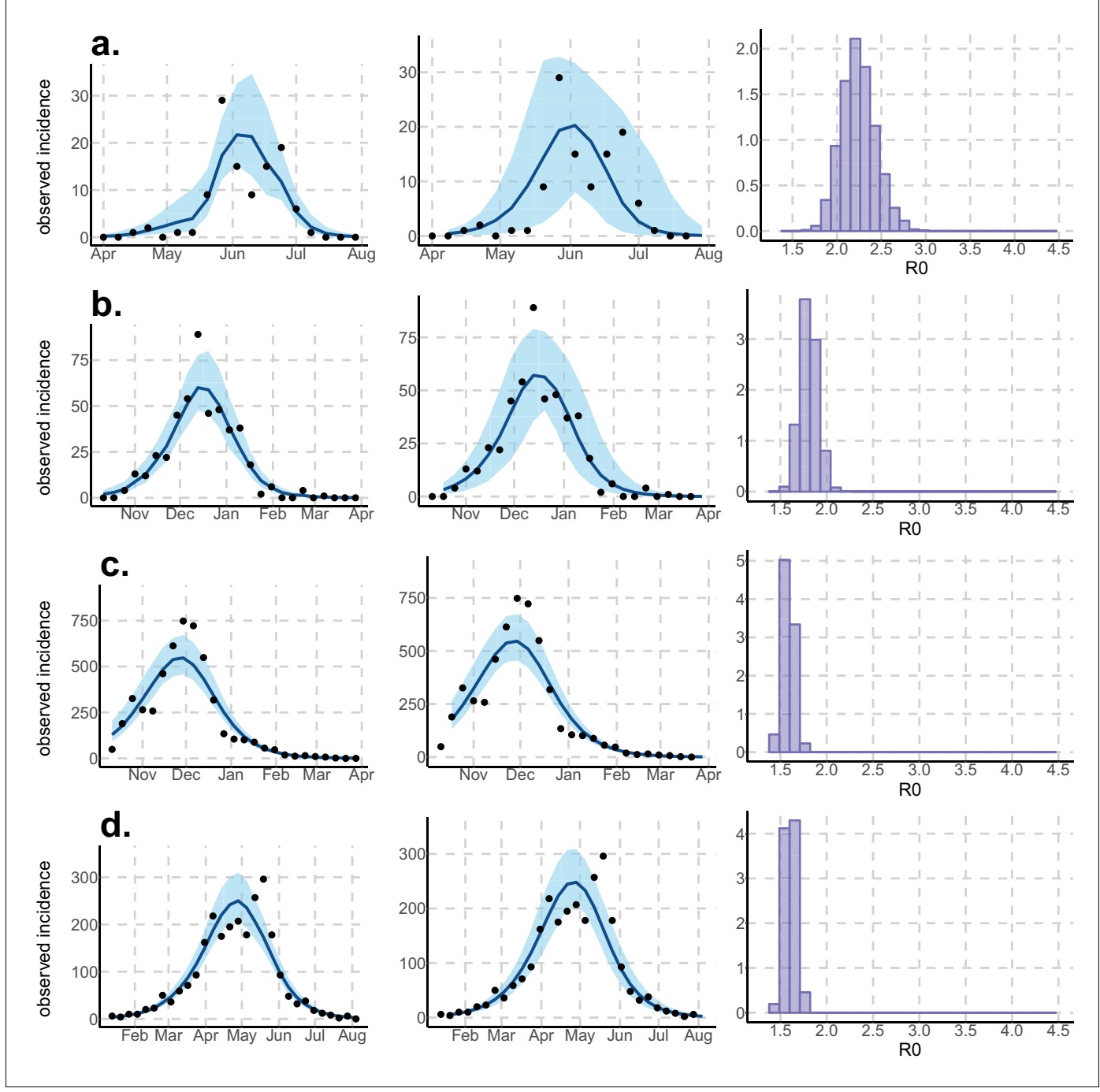

**Figure 3.** Results using the Laneri model. Posterior median number of observed Zika cases (solid line), 95% credible intervals (shaded blue area) and data points (black dots). First column: particle filter fit. Second column: Simulations from the posterior density. Third column: $R_0$ posterior distribution. (a) Yap. (b) Moorea. (c) Tahiti. (d) New Caledonia. The estimated seroprevalences at the end of the epidemic (with 95% credibility intervals) are: (a) 72% (CI95: 68–77, observed 73%); (b) 49% (CI95: 45–53, observed 49%); c) 49% (CI95: 45–53, observed 49%); d) 65% (CI95: 24–91). See *Figure 5*.

involved individuals. In New Caledonia, as no information on seroprevalence was available, the fraction of population involved displays very large confidence intervals (cf. *Tables 1* and *2*), indicating that the model did not manage to identify properly this parameter with the available data. In this case, this parameter is highly correlated to the observation rate $r$ (cf Figures 17 and 21): $r$ and $\rho$ seem unidentifiable without precise information on seroprevalence.

**Table 1.** Parameter estimations for the Pandey model. Posterior median (95% credible intervals). All the posterior parameter distributions are presented in *Figures 6–9* .

| Pandey model | | Yap | Moorea | Tahiti | New Caledonia |
|---|---|---|---|---|---|
| Population size | H | 6892 | 16,200 | 178,100 | 268,767 |
| Basic reproduction number | $R_0$ | 3.2 (2.4–4.1) | 2.6 (2.2–3.3) | 2.4 (2.0–3.2) | 2.0 (1.8–2.2) |
| Observation rate | $r$ | 0.024 (0.019-0.032) | 0.058 (0.048-0.073) | 0.060 (0.050-0.073) | 0.024 (0.010-0.111) |
| Fraction of population involved | $\rho$ | 74% (69–81) | 50% (48–54) | 50% (46–54) | 40% (9–96) |
| Initial number of infected individuals | $H_I(0)$ | 2 (1–8) | 5 (0–21) | 329 (16–1047) | 37 (1–386) |
| Infectious period in human (days) | $\gamma^{-1}$ | 5.2 (4.1–6.7) | 5.2 (4.1–6.8) | 5.2 (4.1–6.7) | 5.5 (4.2–6.8) |
| Extrinsic incubation period in mosquito (days) | $\tau^{-1}$ | 10.6 (8.7–12.5) | 10.5 (8.6–12.4) | 10.5 (8.6–12.6) | 10.7 (8.9–12.5) |
| Mosquito lifespan (days) | $\mu^{-1}$ | 15.6 (11.7–19.3) | 15.3 (11.4–19.1) | 15.1 (11.2–19.0) | 15.4 (11.6–19.4) |

## Discussion

The reproduction number $R_0$ is a key parameter in epidemiology that characterizes the epidemic dynamics and the initial spread of the pathogen at the start of an outbreak in a susceptible population. $R_0$ can be used to inform public health authorities on the level of risk posed by an infectious disease, vaccination strategy, and the potential effects of control interventions (*Anderson and May, 1982*). In the light of the potential public health crisis generated by the international propagation of ZIKV, characterization of the potential transmissibility of this pathogen is crucial for predicting epidemic size, rate of spread and efficacy of intervention.

Using data from both surveillance systems and seroprevalence surveys in four different geographical settings across the Pacific (*Duffy et al., 2009*; *CHSP, 2014*; *Mallet et al., 2015*; *DASS, 2014*; *Aubry et al., 2015b*), we have estimated the basic reproductive number $R_0$ (see *Figures 2–3* and *Tables 1–2*). Our estimate of $R_0$ obtained by inference based on Particle MCMC (*Andrieu et al., 2010*) has values in the range 1.6 (1.5–1.7) – 3.2 (2.4–4.1). Our $R_0$ estimates vary notably across settings. Lower and finer $R_0$ values are found in larger islands. This phenomenon can at least in part be explained by large spatial heterogeneities and higher demographic stochasticity for islands with smaller populations, as well as the influence of stochasticity on biological and epidemiological processes linked to virus transmission. This phenomenon can also be specific to the selection of the studied islands or can reflect a highly clustered geographical pattern, the global incidence curve being the smoothed overview of a collection of more explosive small size outbreaks. However, it is notable that the two French Polynesian islands yield similar estimates of $R_0$ despite differing population sizes. Indeed, other important factors differ among French Polynesia, New Caledonia and Yap, such as the human genetic background and their immunological history linked to the circulation of others arboviruses. Moreover, whilst both New Caledonia and French Polynesia populations were infected by the same ZIKV lineage and transmitted by the same principle vector species, *Aedes aegypti*, the epidemic in Yap occurred

**Table 2.** Parameter estimations for the Laneri model. Posterior median (95% credible intervals). All the posterior parameter distributions are presented in *Figures 10–13*.

| Laneri model | | Yap | Moorea | Tahiti | New Caledonia |
|---|---|---|---|---|---|
| Population size | H | 6892 | 16,200 | 178,100 | 268,767 |
| Basic reproduction number | $R_0$ | 2.2 (1.9–2.6) | 1.8 (1.6–2.0) | 1.6 (1.5–1.7) | 1.6 (1.5–1.7) |
| Observation rate | $r$ | 0.024 (0.019–0.033) | 0.057 (0.047–0.07) | 0.057 (0.049–0.069) | 0.014 (0.010–0.037) |
| Fraction of population involved | $\rho$ | 73% (69–78) | 51% (47–55) | 54% (49–59) | 71% (27–98) |
| Initial number of infected individuals | $H_I(0)$ | 2 (1–10) | 9 (1–28) | 667 (22–1570) | 82 (2–336) |
| Infectious period in human (days) | $\gamma^{-1}$ | 5.3 (4.1–6.6) | 5.3 (4.1–6.7) | 5.2 (4.1–6.7) | 5.4 (4.1–6.8) |
| Extrinsic incubation period in mosquito (days) | $\tau^{-1}$ | 10.7 (8.8–12.7) | 10.6 (8.6–12.6) | 10.5 (8.5–12.5) | 10.8 (8.9–12.8) |

much earlier with a different ZIKV lineage (*Wang et al., 2016*) and vectored by a different mosquito species *Aedes hensilli* (*Ledermann et al., 2014*). In French Polynesia, the vector *Aedes polynesiensis* is also present and dominates in Moorea with higher densities than in Tahiti. Finally, different vector control measures have been conducted in the three countries.

To date, studies investigating Zika outbreaks in the Pacific have always estimated $R_0$ using a deterministic framework. Using a similar version of the Pandey model in French Polynesia, Kucharski et al. (*Kucharski et al., 2016*) estimated $R_0$ between 1.6 and 2.3 (after scaling to square root for comparison) for Tahiti and between 1.8 and 2.9 in Moorea. These estimates are slightly lower and less variable than ours. This difference can be explained firstly by the chosen priors on mosquito parameters and secondly because our model includes demographic stochasticity. Moreover, they predicted a seroprevalence rate at the end of the epidemic of 95–97%, far from the 49% measured. In Yap island, a study (*Funk et al., 2016*) used a very detailed deterministic mosquito model, and estimated an $R_0$ for Zika between 2.9 and 8. In this case, our lower and less variable estimates may come from the fact that our model is more parsimonious in the number of uncertain parameters, especially concerning the mosquito population. Finally, a third study (*Nishiura et al., 2016a*) relied on another method for $R_0$ calculation (based on the early exponential growth rate of the epidemic) in French Polynesia as a whole and in Yap. Again, the obtained parameters are lower than ours in French Polynesia and higher in Yap. The first estimations for South America using a similar methodology (*Nishiura et al., 2016b*; *Towers et al., 2016*; *Gao et al., 2016*) lead to similar $R_0$ values. In all these studies a deterministic framework is used excluding the possibility of accounting for the high variability of biological and epidemiological processes exacerbated by the small size of the population. In these three studies, like in ours, it is worth noting that little insight is obtained regarding mosquito parameters. Posterior distribution mimics the chosen prior (cf. Figures 6–13). Both the simulation of the epidemics and the estimated $R_0$ are highly sensitive to the choice of priors on mosquito parameters, for which precise field measures are rare.

In the absence of sufficient data, the modeling of mosquito-borne pathogen transmission is a difficult task due to non-linearity and non-stationarity of the involved processes (*Cazelles and Hales, 2006*). This work has then several limitations. First, our study is limited by the completeness and quality of the data, with regard to both incidence and seroprevalence, but, above all, by the scarcity of information available on mosquitoes. Incidence data is aggregated at the island scale and cannot disentangle the effects of geography and observation noise to explain bimodal curves observed in Yap and New Caledonia. Moreover, although all data came from national surveillance systems, we had very little information about the potential degree of under-reporting, especially due to the high proportion of mildly symptomatic cases, at a time when the dangerous complications associated with the virus were unknown. Moreover, some cases might have been misdiagnosed as other flaviviruses, due to similarity in clinical manifestations or cross-reactivity in assays. Seroprevalence data were gathered from small sample sizes and were also sensitive to cross reactivity in populations non naive to dengue. In addition, they were missing in New Caledonia, which leads to strong correlation between our estimation of the observation rate and the fraction of the population involved in the epidemic. Because of the high proportion of asymptomatic or mildly symptomatic cases, the magnitude of the outbreaks is difficult to evaluate without precise seroprevalence data (*Metcalf et al., 2016*) or detection of mild, asymptomatic or pre-symptomatic infections (*Thompson et al., 2016*). Considering vectors, no demographic data were available and this partly explains the large variability of our $R_0$ estimations. Secondly, incidence and seroprevalence data were difficult to reconcile; the use of incidence data led to higher infection rates than those observed in the seroprevalence data. This difficulty has been overcome by considering that only a fraction of the population ($\rho$) is involved in the epidemic and then our model manages to reproduce the observed seroprevalence rate. This exposed fraction could be the result of spatial heterogeneity and high clustering of cases and transmission, as observed for dengue. The small dispersal of the mosquito and the limited population inter-mingling likely leads to considerable spatial variation in the extent of exposure to the virus and pockets of refugia in Tahiti as elsewhere (*Telle et al., 2016*). For instance, previous dengue infection rates in French Polynesia display large spatial variations even within islands (*Daudens et al., 2009*). Finer scale incidence and seroprevalence data would be useful to explore this. Another explanation for higher predicted than observed infection rates could be due to interaction with other flaviviruses.

The Zika outbreak was concomitant with dengue outbreaks in French Polynesia (*CHSP, 2014*; *Mallet et al., 2015*) and New Caledonia (*DASS, 2014*). Examples of coinfection have been reported (*Dupont-Rouzeyrol et al., 2015*) but competition between these close pathogens may also have occurred. Finally, mathematical models with vectorial transmission may tend to estimate high attack rates, sometimes leading to a contradiction between observed incidence and observed seroprevalence. Assumptions on the proportionality between infected mosquitoes and the force of infection, as well as the density-dependence assumption in these models could be questioned. Indeed even if these assumptions are at the heart of the mathematical models of mosquito-borne pathogen transmission (*Reiner et al., 2013*; *Smith et al., 2014*), a recent review (*Halstead, 2008*) and recent experimental results (*Bowman et al., 2014*; *Harrington et al., 2014*) question these important points.

On a final note, the estimates of $R_0$ for ZIKV are similar to but generally on the lower side of estimates made for two other flaviviruses of medical importance, dengue and Yellow Fever viruses (*Favier et al., 2006*; *Imai et al., 2015*; *Massad et al., 2003*), even though caution is needed in the comparison of studies with differing models, methods and data sources. Interventions strategies developed for dengue should thus enable as, if not more effective control of ZIKV, with the caveat that ZIKV remains principally a mosquito-borne pathogen with little epidemiological significance of the sexual transmission route. Even though further work and data are needed to support this hypothesis (*Brauer et al., 2016*), two recent studies indicated that sexual transmission alone is not sufficient to sustain an epidemic (*Gao et al., 2016*; *Towers et al., 2016*).

In conclusion, using recent stochastic modeling methods, we have been able to determine estimates of $R_0$ for ZIKV with an unexpected relationship with population size. Further data from the current Zika epidemic in South America that is caused by the same lineage as French Polynesia lead to estimates in the same range of values (*Nishiura et al., 2016b*; *Towers et al., 2016*; *Gao et al., 2016*). Our study highlights the importance of gathering seroprevalence data, especially for a virus that often leads to an asymptomatic outcome and it would provide a key component for precise quantitative analysis of pathogen propagation to enable improved planning and implementation of prevention and control strategies.

## Materials and methods

### Data

During the 2007 outbreak that struck Yap, 108 suspected or confirmed Zika cases (16 per 1000 inhabitants) were reported by reviewing medical records and conducting prospective surveillance between April 1st and July 29th 2007 (*Duffy et al., 2009*). In French Polynesia, sentinel surveillance recorded more than 8700 suspected cases (32 per 1000 inhabitants) across the whole territory between October 2013 and April 2014 (*CHSP, 2014*; *Mallet et al., 2015*). In New Caledonia, the first Zika case was imported from French Polynesia on 2013 November 12th. Approximately 2500 cases (9 per 1000 inhabitants) were reported through surveillance between January (first autochtonous case) and August 2014 (*DASS, 2014*).

For Yap and French Polynesia, the post-epidemic seroprevalence was assessed. In Yap, a household survey was conducted after the epidemic, yielding an infection rate in the island of 73% (*Duffy et al., 2009*). In French Polynesia, three seroprevalence studies were conducted. The first one took place before the Zika outbreak, and concluded that most of the population was naive for Zika virus (*Aubry et al., 2015a*). The second seroprevalence survey was conducted between February and March 2014, at the end of the outbreak, and reported a seroprevalence rate around 49% (*Aubry et al., 2015b*). The third one concerned only schoolchildren in Tahiti and was therefore not included in the present study.

Demographic data on population size were based on censuses from Yap (*Duffy et al., 2009*), French Polynesia (*Insee, 2012*), and New Caledonia (*Insee, 2014*).

### Models and inference
#### Model equations
Although the models are simulated in a stochastic framework, we present them with ordinary differential equations for clarity. The reactions involved in the stochastic models are the same as

those governed by the deterministic equations, but the simulation process differs through the use of discrete compartments. It is described in the next section.

The equations describing Pandey model are:

$$\frac{dH_S}{dt} = -\beta_H v_I H_S$$
$$\frac{dH_E}{dt} = \beta_H v_I H_S - \sigma H_E$$
$$\frac{dH_I}{dt} = \sigma H_E - \gamma H_I$$
$$\frac{dH_R}{dt} = \gamma H_I$$
$$\frac{dv_S}{dt} = \mu - \frac{\beta_V H_I}{N} v_S - \mu v_S$$
$$\frac{dv_E}{dt} = \frac{\beta_V H_I}{N} v_S - \tau v_E - \mu v_E$$
$$\frac{dv_I}{dt} = \tau v_E - \mu v_I$$

where $v_s = \frac{V_S}{V}$ is the proportion of susceptible mosquitoes, $v_E = \frac{V_E}{V}$ the proportion of exposed mosquitoes, and $v_I = \frac{V_I}{V}$ the proportion of infected mosquitoes. Since we are using a discrete model, we cannot use directly the proportions $v_S$, $v_E$ and $v_I$ whose values are smaller than one. Therefore, we rescale using $V = H$, which leads to $V_S' = v_S \cdot H$, $V_E' = v_E \cdot H$, and $V_I' = v_I \cdot H$. In this model, the force of infection for humans is $\lambda_H = \beta_H v_I$, and the force of infection for mosquitoes is $\lambda_V = \beta_V \frac{H_I}{N}$

The equations describing Laneri model are:

$$\frac{dH_S}{dt} = -\lambda H_S$$
$$\frac{dH_E}{dt} = \lambda H_S - \sigma H_E$$
$$\frac{dH_I}{dt} = \sigma H_E - \gamma H_I$$
$$\frac{dH_R}{dt} = \gamma H_I$$
$$\frac{d\kappa}{dt} = \frac{2\beta H_I \tau}{N} - 2\tau\kappa$$
$$\frac{d\lambda}{dt} = 2\tau\kappa - 2\tau\lambda$$

In this model, the role of mosquitoes in transmission is represented only through the delay they introduce during the extrinsic incubation period (EIP, incubation period in the mosquito). For modeling reasons, this delay is included by representing the force of infection from infected humans to susceptible humans with two compartments $\kappa$ and $\lambda$: in this formalism, the duration between the moment when an exposed individual becomes infectious and the moment when another susceptible individual acquires the infection has a gamma distribution of mean $\tau^{-1}$(*Laneri et al., 2010*; *Roy et al., 2013*; *Lloyd, 2001*). Therefore, $\lambda$ represents the current force of infection for humans $\lambda_H = \lambda$ . The compartment $\kappa$ represents the same force of infection but at a previous stage, reflecting the exposed phase for mosquitoes during the extrinsic incubation period. As an analogy to Pandey model, the force of infection for mosquitoes is $\lambda_V = \frac{\beta H_I \tau}{v_s N}$, and therefore, the parameter $\beta$ can be interpreted as the product of a transmission parameter $\beta'$ by the proportion of susceptible mosquitoes: $\beta = v_s \beta'$. The force of infection for mosquitoes is then similar to Pandey's : $\lambda_V = \beta' \tau \frac{H_I}{N}$.

Again, since we are using a discrete model, we cannot use directly the proportions $\lambda$ and $\kappa$ whose values are smaller than one. Therefore, we rescale up to a factor $N$, which leads to $L = \lambda N$ and $K = \kappa N$.

In both models, following the dominant paradigm that diseases transmitted by *Aedes* mosquitoes are highly clustered, we restricted the total population $H$ measured by census to a fraction $N = \rho.H$, in which the parameter $\rho$ is estimated. This formulation makes the hypothesis that a fraction of the total population is not at risk from the epidemic, because of individual factors or because the individuals remain in areas where the virus is not present. Moreover as the vector's flying range is small, the clustering of ZIKV infection may be reinforced. This may result in escapees from infection within the population, even at a single island scale. The available data does not allow further exploration of

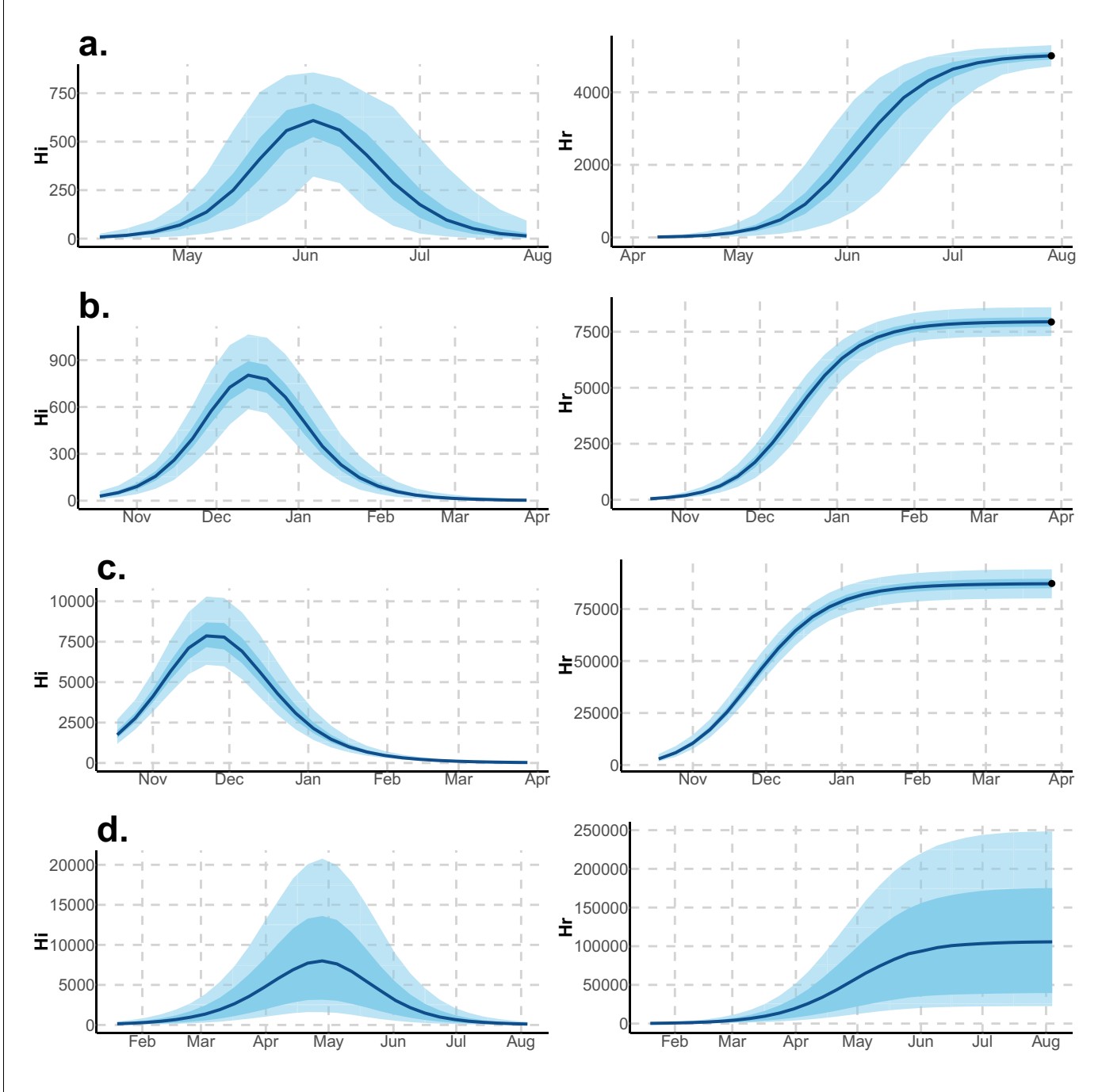

**Figure 4.** Infected and recovered humans evolution during the outbreak with Pandey model. Simulations from the posterior density: posterior median (solid line), 95% and 50% credible intervals (shaded blue areas) and observed seroprevalence (black dots). First column: Infected humans ($H_I$). Second column: Recovered humans ($H_R$). (a) Yap. (b) Moorea. (c) Tahiti. (d) New Caledonia.

the mechanisms underlying these phenomena, which seem fundamental to understand ZIKV propagation. At the very least, the restriction to a fraction $\rho$ enables the model to reproduce the observed seroprevalence rates, and to provide coherent results with respect to both data sources (seroprevalence and surveillance data).

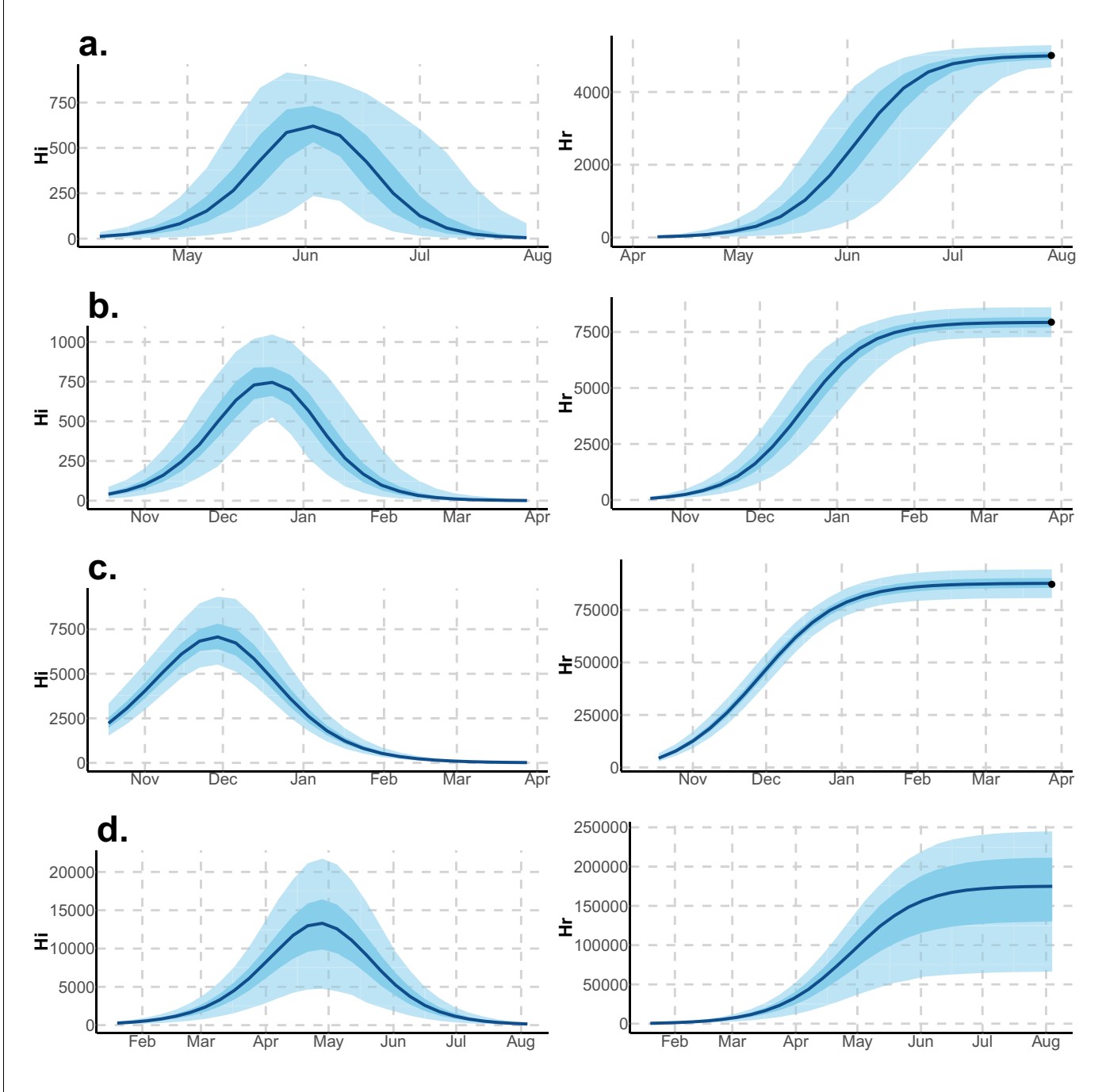

**Figure 5.** Infected and recovered humans evolution during the outbreak with Laneri model. Simulations from the posterior density: posterior median (solid line), 95% and 50% credible intervals (shaded blue areas) and observed seroprevalence (black dots). First column: Infected humans ($H_I$). Second column: Recovered humans ($H_R$). (a) Yap. (b) Moorea. (c) Tahiti. (d) New Caledonia.

## Stochastic framework

Both models are simulated in a stochastic and discrete framework, the Poisson with stochastic rates formulation (*Bretó et al., 2009*), to include the uncertainties related to small population size. In this framework, the number of reactions occurring in a time interval $dt$ is approximated by a multinomial distribution. In a model with $m$ possible reactions and $c$ compartments, $z_t$ being the state of the

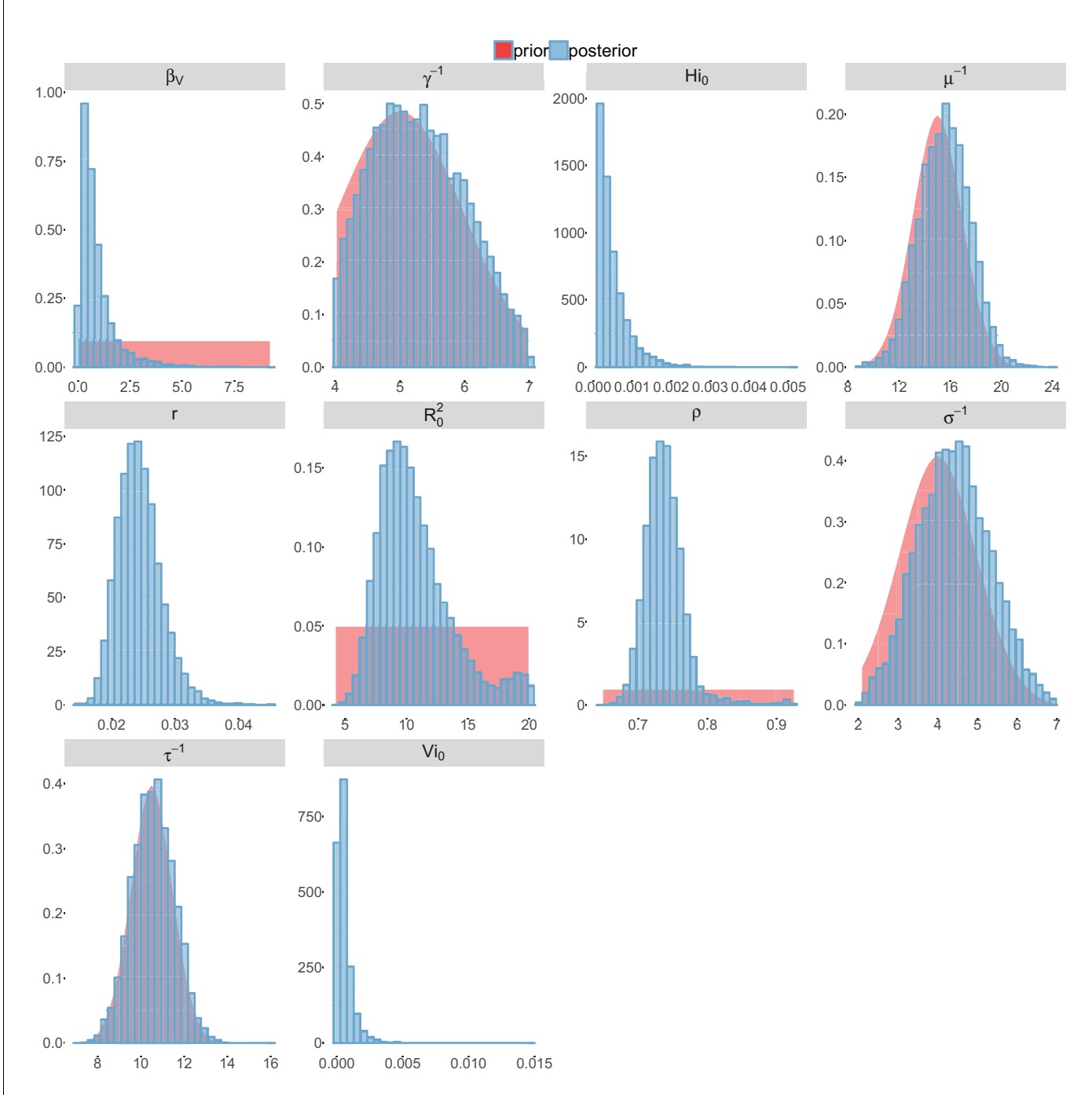

**Figure 6.** Posterior distributions. Pandey model, Yap island.

system at time $t$ and $\theta$ the model parameters, the probability that each reaction with rate $r^k$ occurs $n_k$ times in $dt$ is calculated as follows (*Dureau et al., 2013*):

$$p(n_1, \ldots n_m | z_t, \theta) = \prod_{i=1}^{c} \left\{ M_i \left(1 - \sum_{X(k)=i} p_k\right)^{\bar{n}_i} \prod_{X(k)=i} (p_k)^{n_k} \right\} + o(dt)$$

with, $z_t^{(i)}$ being the number of individual in compartment $i$ at time $t$,

- $p_k = \left(1 - exp\left\{-\sum_{X(l)=i} r^l(z_t, \theta) z_t^{X(l)} dt\right\}\right) \frac{r^k(z_t, \theta)}{\sum_{X(l)=i} r^l(z_t, \theta)}$

- $\bar{n}_i = z_t^{(i)} - \sum_{X(k)=i} n_k$ (the number of individuals staying in compartment $i$ in $dt$)

- $M_i = \binom{z_t^{(i)}}{n_{k X(k)=i} \bar{n}_i}$ (multinomial coefficient)

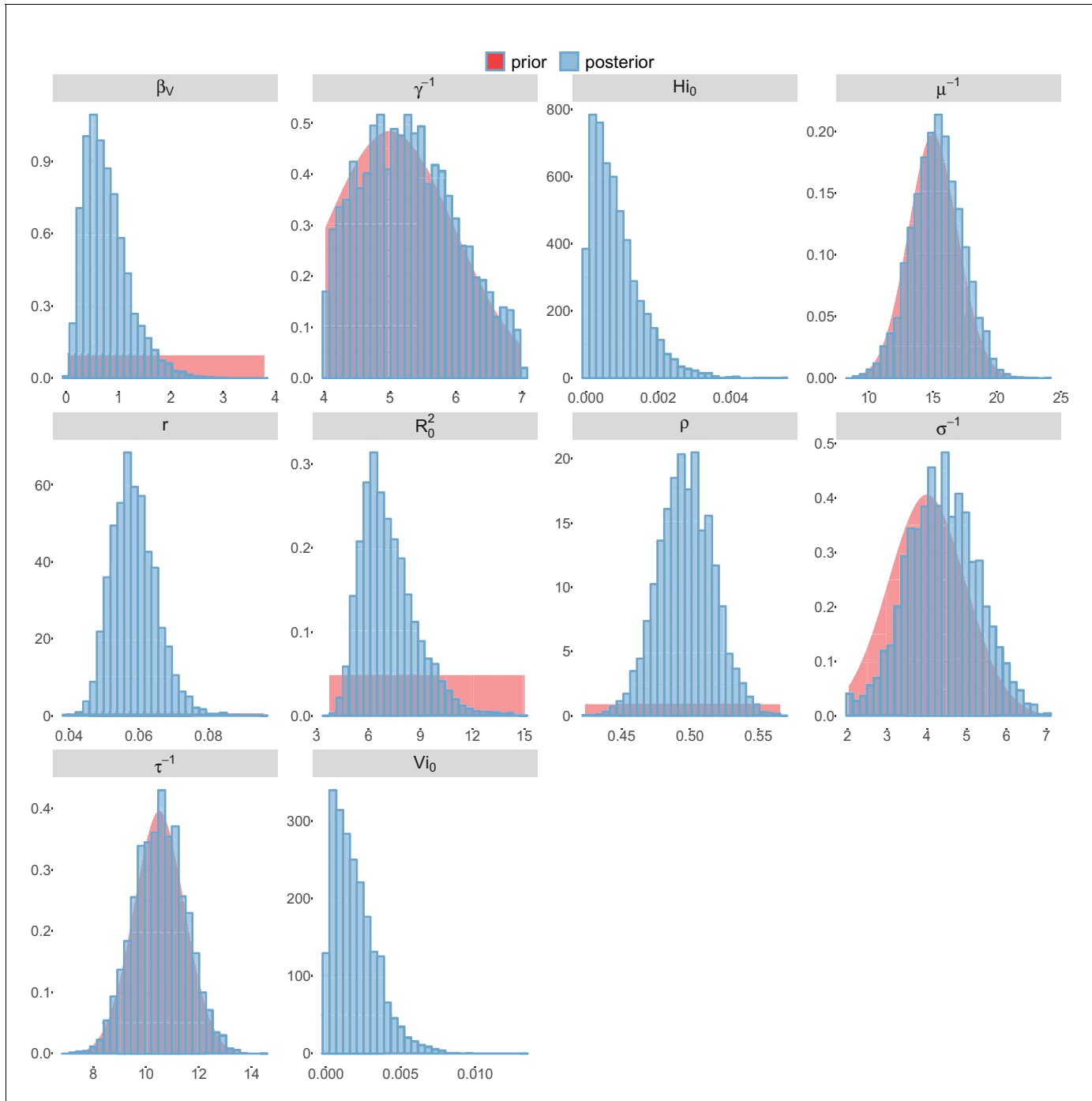

**Figure 7.** Posterior distributions. Pandey model, Moorea island.

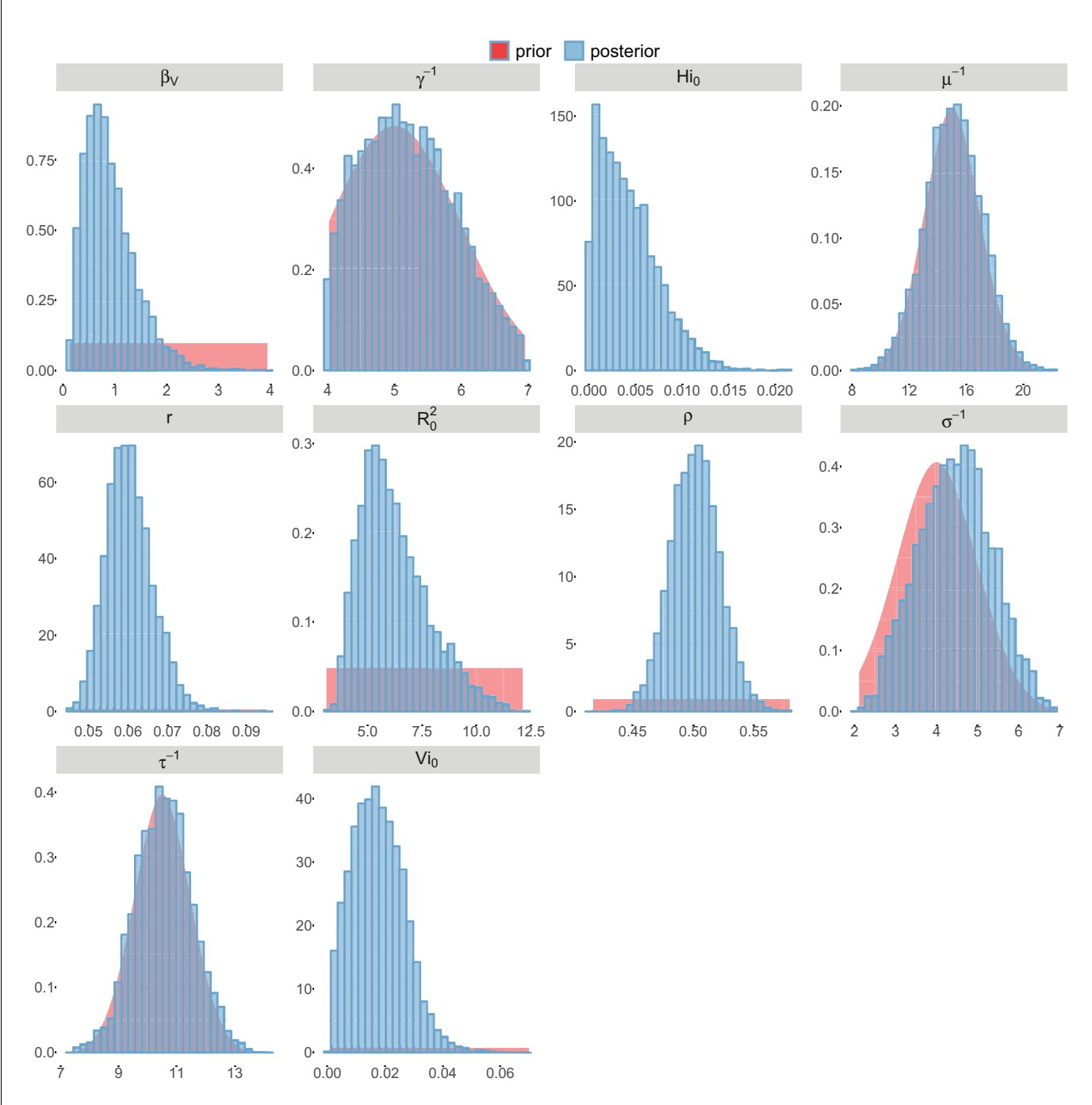

**Figure 8.** Posterior distributions. Pandey model, Tahiti island.

## Observation models

The only observed compartments are the infected humans (incidence measured every week) and the recovered humans (seroprevalence at the end of the outbreak when data is available). In order to link the model to the data, two observation models, for both incidence and seroprevalence data, are needed.

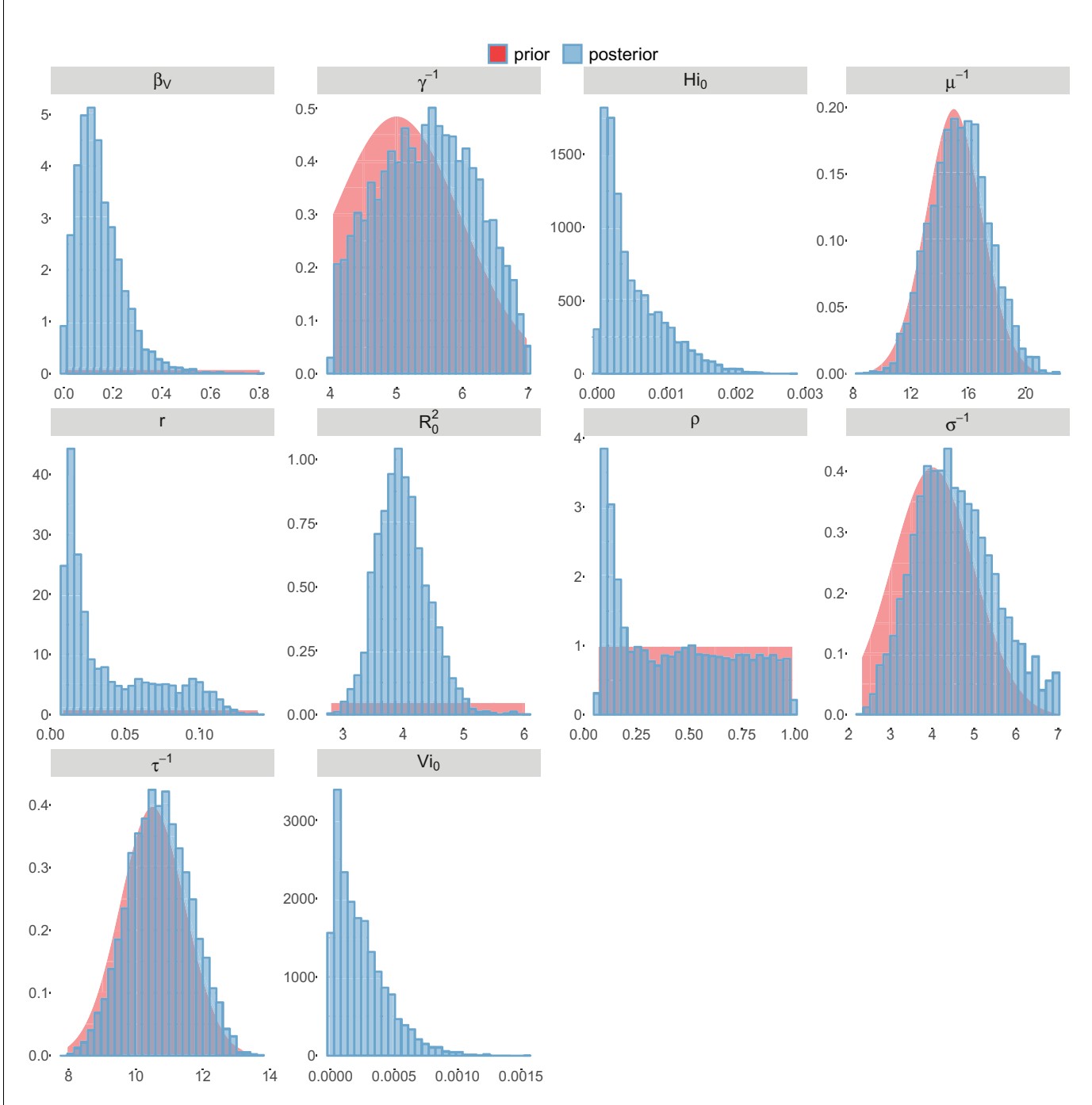

**Figure 9.** Posterior distributions. Pandey model, New Caledonia.

## Observation model on incidence data

The observed weekly incidence is assumed to follow a negative binomial distribution (*Bretó et al., 2009*) whose mean equals the number of new cases predicted by the model times an estimated observation rate $r$.

The observation rate $r$ accounts for non observed cases, due to non reporting from medical centers, mild symptoms unseen by health system, and asymptomatic infections. Without additional data, it is not possible to make a distinction between these three categories of cases. We also

implicitely make the assumption that these cases transmit the disease as much as reported symptomatic cases.

The observation model for incidence data is therefore :

$$Inc_{obs} = NegBin(\phi^{-1}, \frac{1}{1 + \phi rInc})$$

$Inc_{obs}$ being the observed incidence, and $Inc$ the incidence predicted by the model. The dispersion parameter (*Bretó et al., 2009*) $\phi$ is fixed at 0.1.

## Observation model on seroprevalence data

Seroprevalence data is fitted for Tahiti, Moorea, and Yap settings. It is assumed that the observed seroprevalence at the end of the epidemic follows a normal distribution with fixed standard deviation, whose mean equals the number of individuals in the $H_R$ compartment predicted by the model.

The observation model for seroprevalence data is therefore :

$$H_R^{obs} = Normal(H_R, \Lambda)$$

at the last time step, with notations detailed for each model in *Table 3*.

## Prior distributions

Informative prior distributions are assumed for the mosquito lifespan, the duration of infectious period, and both intrinsic and extrinsic incubation periods. The initial numbers of infected mosquitoes and humans are estimated, and the initial number of exposed individuals is set to the initial number of infected to reduce parameter space. We assume that involved populations are naive to Zika virus prior to the epidemic and set the initial number of recovered humans to zero. The other priors and associated references are listed in *Table 4*.

The range for the prior on observation rate is reduced for Tahiti and New Caledonia, in order to reduce the parameter space and facilitate convergence. In both cases, we use the information provided with the data source. In French Polynesia, 8750 cases we reported, but according to local health authorities, more than 32,000 people would have attended health facilities for Zika (*Mallet et al., 2015*) (8750/32000 $\leq$ 0.3). In New Caledonia, approximately 2500 cases were reported but more than 11,000 cases were estimated by heath authorities (*DASS, 2014*) (2500/11000 $\leq$ 0.23). In both cases, these extrapolations are lower bounds on the real number of cases (in particular, they do not estimate the number of asymptomatic infections), and therefore can be used as upper bounds on the observation rate.

## Estimations

### Inference with PMCMC

The complete model is represented using the state space framework, with two equation systems: the transition equations refer to the transmission models, and the measurement equations are given by the observation models.

In a deterministic framework, this model could be directly estimated using MCMC, with a Metropolis-Hastings algorithm targeting the posterior distribution of the parameters. This algorithm would require the calculation of the model likelihood at each iteration.

**Table 3.** Details of the observation models for seroprevalence

| Island | Date | Standard deviation | Observed seroprevalence |
|--------|------|--------------------|--------------------------|
| | | $\Lambda$ | $H_R^{obs}$ |
| Yap | 2007-07-29 | 150 | 5005 (*Duffy et al., 2009*) |
| Moorea | 2014-03-28 | 325 | 0.49 × 16200 = 7938 (*Aubry et al., 2015b*) |
| Tahiti | 2014-03-28 | 3562 | 0.49 × 178100 = 87269 (*Aubry et al., 2015b*) |

**Table 4.** Prior distributions of parameters. 'Uniform[0,20]' indicates a uniform distribution in the range [0,20]. 'Normal(5,1) in [4,7]' indicates a normal distribution with mean five and standard deviation 1, restricted to the range [4,7].

| Parameters | | Pandey model | Laneri model | References |
|---|---|---|---|---|
| $R_0^2$ | squared basic reproduction number | Uniform[0, 20] | Uniform[0, 20] | assumed |
| $\beta_V$ | transmission from human to mosquito | Uniform[0,10] | . | assumed |
| $\gamma^{-1}$ | infectious period (days) | Normal(5,1) in [4,7] | Normal(5,1) in [4,7] | (*Mallet et al., 2015*) |
| $\sigma^{-1}$ | intrinsic incubation period (days) | Normal(4,1) in [2,7] | Normal(4,1) in [2,7] | (*Nishiura et al., 2016b*; *Bearcroft, 1956*; *Lessler et al., 2016*) |
| $\tau^{-1}$ | extrinsic incubation period (days) | Normal(10.5,1) in [4,20] | Normal(10.5,1) in [4,20] | (*Hayes, 2009*; *Chouin-Carneiro et al., 2016*) |
| $\mu^{-1}$ | mosquito lifespan (days) | Normal(15,2) in [4,30] | . | (*Brady et al., 2013*; *Liu-Helmersson et al., 2014*) |
| $\rho$ | fraction of population involved | Uniform[0,1] | Uniform[0,1] | |
| **Initial conditions (t=0)** | | **Pandey model** | **Laneri model** | |
| $H_I(0)$ | infected humans | Uniform[$10^{-6}$,1]N | Uniform[$10^{-6}$,1]N | |
| $H_E(0)$ | exposed humans | $H_I(0)$ | $H_I(0)$ | |
| $H_R(0)$ | recovered humans | 0 | 0 | |
| | infected vectors | $V_I(0)$=Uniform[$10^{-6}$,1]H | $L(0)$=Uniform[$10^{-6}$,1]N | |
| | exposed vectors | $V_E(0) = V_I(0)$ | $K(0)=L(0)$ | |

| Local conditions | | Yap | Moorea | Tahiti | New Caledonia | References |
|---|---|---|---|---|---|---|
| r | observation rate | Uniform [0,1] | Uniform [0,1] | Uniform [0,0.3] | Uniform [0,0.23] | (*Mallet et al., 2015*; *DASS, 2014*) |
| H | population size | 6,892 | 16,200 | 178,100 | 268,767 | (*Duffy et al., 2009*; *Insee, 2012, 2014*) |

In our stochastic framework, the model output is given only through simulations and the likelihood is intractable. In consequence, estimations are performed with the PMCMC algorithm (particle Markov Chain Monte Carlo (*Andrieu et al., 2010*)), in the PMMH version (particle marginal Metropolis-Hastings). This algorithm uses the Metropolis-Hastings structure, but replaces the real likelihood by its estimation with Sequential Monte Carlo (SMC).

Algorithm 1 PMCMC (*Andrieu et al., 2010*) (PMMH version, as in SSM (*Dureau et al., 2013*))

In a model with $n$ observations and $J$ particles.
$q(.|\theta^{(i)})$ is the transition kernel.

1: Initialize $\theta^{(0)}$.
2: Using SMC algorithm, compute $\hat{p}(y_{1:n}|\theta^{(0)})$ and sample $x_{0:n}^*$ from $\hat{p}(x_{0:n}|y_{1:n},\theta^{(0)})$.
3: **for** $i = 1...N$ **do**
4:   Sample $\theta^*$ from $q(.|\theta^{(i)})$
5:   Using SMC algorithm, compute $L(\theta^*) = \hat{p}(y_{1:n}|\theta^*)$ and sample $x_{0:n}^*$ from $\hat{p}(x_{0:n}|y_{1:n},\theta^*)$
6:   Accept $\theta^*$ (et $x_{0:n}^*$) with probability $1 \wedge \frac{L(\theta^{(i)})q(\theta^*)}{L(\theta^*)q(\theta^*|\theta^{(i)})}$
7:   If accepted, $\theta^{(i+1)} = \theta^*$ and $x_{0:n}^{(i+1)} = x_{0:n}^*$. Otherwise $\theta^{(i+1)} = \theta^{(i)}$ and $x_{0:n}^{(i+1)} = x_{0:n}^{(i)}$.
8: **end for**

SMC (*Doucet et al., 2001*) is a filtering method that enables to recover the latent variables and estimate the likelihood for a given set of parameters. The data is treated sequentially, by adding one more data point at each iteration. The initial distribution of the state variables is approximated by a sample a particles, and from one iteration to the next, all the particles are projected according

to the dynamic given by the model. The particles receive a weight according to the quality of their prediction regarding the observations. Before the next iteration, all the particles are resampled using these weights, in order to eliminate low weight particles and concentrate the computational effort in high probability regions. Model likelihood is also computed sequentially at each iteration (*Dureau et al., 2013*; *Doucet and Johansen, 2011*).

---

**Algorithm 2 SMC** (Sequential Monte Carlo, as implemented in SSM [*Dureau et al., 2013*])

In a model with $n$ observations and $J$ particles.

$L$ is the model likelihood $p(y_{1:T}|\theta)$. $W_k^{(j)}$ is the weight and $x_k^{(j)}$ is the state associated to particle $j$ at iteration $k$.

1: Set $L = 1$, $W_0^{(j)} = 1/J$.
2: Sample $(x_0^{(j)})_{j=1:J}$ from $p(x|\theta_0)$.
3: **for** $k = 0 : n - 1$ **do**
4:    **for** $j = 0 : j$ **do**
5:       Sample $(x_{k+1}^{(j)})_{j=1:J}$ from $p(x_{k+1}|x_k, \theta)$
6:       Set $\alpha^{(j)} = p(y_{k+1}|x_{k+1}^{(j)}, \theta)$
7:    **end for**
8:    Set $W_{k+1}^{(j)} = \frac{\alpha^{(j)}}{\sum_{l=1}^{J} \alpha^{(l)}}$ and $L = L\frac{1}{J}\sum_j \alpha^{(l)}$
9:    Resample $(x_{0:k+1}^{(j)})_{j=1:J}$ from $W_{k+1}^{(j)}$
10: **end for**

---

A gaussian kernel $q(.|\theta^{(i)})$ is used in the PMCMC algorithm, with mean $\theta^{(i)}$ and fixed variance $\Sigma^q$ (random walk Metropolis Hastings).

## Initialization

PMCMC algorithm is very sensitive to initialization of both the parameter values $\theta^{(0)}$ and the covariance matrix $\Sigma^q$. Several steps of initialization are therefore used.

Firstly, parameter values are initialized by maximum likelihood through simplex algorithm on a deterministic version of the model. We apply the simplex algorithm to a set of 1000 points sampled in the prior distributions and we select the parameter set with the highest likelihood.

Secondly, in order to initialise the covariance matrix, an adaptive MCMC (Metropolis Hastings) framework is used (*Roberts and Rosenthal, 2009*; *Dureau et al., 2013*). It uses the empirical covariance of the chain $\Sigma^{(i)}$, and aims to calibrate the acceptance rate of the algorithm to an optimal value. The transition kernel is also mixed (with a probability $\alpha = 0.05$) with another gaussian using the identity matrix to improve mixing properties.

$$q^A(.|x^{(i)}) = \alpha N(x^{(i)}, \lambda\frac{2.38^2}{d}Id) + (1 - \alpha)N(x^{(i)}, \lambda\frac{2.38^2}{d}\Sigma^{(i)})$$

The parameter $\lambda$ is approximated by successive iterations using the empirical acceptance rate of the chain.

$$\lambda_{i+1} = \lambda_i a^i(AccRate_i - 0.234)$$

The adaptative PMCMC algorithm itself may have poor mixing properties without initialization. A first estimation of the covariance matrix is computed using KMCMC algorithm (*Dureau et al., 2013*). In the KMCMC algorithm, the model is simulated with stochastic differential equations (intermediate between deterministic and Poisson with stochastic rates frameworks) and the SMC part of the adaptative PMCMC is replaced by the extended Kalman filter. When convergence is reached with KMCMC, then, adaptative PMCMC is used.

The PMCMC algorithm is finally applied on the output of the adaptative PMCMC, using 50,000 iterations and 10,000 particles. Calculations are performed with SSM software (*Dureau et al., 2013*) and R version 3.2.2.

## $R_0$ Calculation

$R_0$ is calculated using the Next Generation Matrix approach (NGM) (19).

## $R_0$ Calculation in Pandey model

$$F = \begin{pmatrix} 0 & 0 & 0 & \beta_H \\ 0 & 0 & 0 & 0 \\ 0 & \beta_v & 0 & 0 \\ 0 & 0 & 0 & 0 \end{pmatrix} \qquad v = \begin{pmatrix} -\sigma & 0 & 0 & 0 \\ 0 & -\gamma & 0 & 0 \\ 0 & 0 & -(\mu+\tau) & 0 \\ 0 & 0 & 0 & -\mu \end{pmatrix}$$

Then we have,

$$V^{-1} = \begin{pmatrix} -1/\sigma & 0 & 0 & 0 \\ -1/\gamma & -1/\gamma & 0 & 0 \\ 0 & 0 & -1/(\mu+\tau) & 0 \\ 0 & 0 & -\tau/[\mu(\tau+\mu)] & -1/\mu \end{pmatrix}$$

and

$$FV^{-1} = \begin{pmatrix} 0 & 0 & -\beta_H\tau/[\mu(\tau+\mu)] & -\beta_H/\mu \\ 0 & 0 & 0 & 0 \\ -\beta_v/\gamma & -\beta_v/\gamma & 0 & 0 \\ 0 & 0 & 0 & 0 \end{pmatrix}$$

We calculate the eigen values $\alpha$ of $-FV^{-1}$ :

$$\begin{vmatrix} -\alpha & 0 & \beta_H\tau/[\mu(\tau+\mu)] & \beta_H/\mu \\ 0 & -\alpha & 0 & 0 \\ \beta_v/\gamma & \beta_v/\gamma & -\alpha & 0 \\ 0 & 0 & 0 & -\alpha \end{vmatrix} = \alpha^2\left(\alpha^2 - \frac{\beta_H\beta_V\tau}{\gamma\mu(\tau+\mu)}\right) = 0$$

Then $\alpha = 0$ or $\alpha = \pm\sqrt{\frac{\beta_H\beta_V\tau}{\gamma\mu(\tau+\mu)}}$ and the highest eigenvalue is $R_0 = \sqrt{\frac{\beta_H\beta_V\tau}{\gamma\mu(\tau+\mu)}}$.

This formula defines $R_0$ as "the number of secondary cases per generation" (**Dietz, 1993**): i.e $R_0$ can be written as the geometric mean $R_0 = \sqrt{R_0^v R_0^h}$, where $R_0^v$ is the number of infected mosquitoes after the introduction of one infected human in a naive population, and $R_0^h$ is the number of infected humans after the introduction of one infected mosquito in a naive population. With this definition, herd immunity is reached when $(1 - R_0^{-2})$ of the population is vaccinated (**Dietz, 1993**).

## $R_0$ Calculation in Laneri model
Following the analogy with Pandey model, we compute the spectral radius of the NGM for the Laneri model.

$$F = \begin{pmatrix} 0 & 0 & 0 & 1 \\ 0 & 0 & 0 & 0 \\ 0 & \beta\tau & 0 & 0 \\ 0 & 0 & 0 & 0 \end{pmatrix} \qquad V = \begin{pmatrix} -\sigma & 0 & 0 & 0 \\ 0 & -\gamma & 0 & 0 \\ 0 & 0 & -\tau & 0 \\ 0 & 0 & \tau & -\tau \end{pmatrix}$$

Then we have,

$$V^{-1} = \begin{pmatrix} -1/\sigma & 0 & 0 & 0 \\ -1/\gamma & -1/\gamma & 0 & 0 \\ 0 & 0 & -1/\tau & 0 \\ 0 & 0 & -1/\tau & -1/\tau \end{pmatrix}$$

and

$$FV^{-1} = \begin{pmatrix} 0 & 0 & -1/\tau & -1/\tau \\ 0 & 0 & 0 & 0 \\ -\beta\tau/\gamma & -\beta\tau/\gamma & 0 & 0 \\ 0 & 0 & 0 & 0 \end{pmatrix}$$

We calculate the eigen values $\alpha$ of $-FV^{-1}$ :

$$\begin{vmatrix} -\alpha & 0 & 1/\tau & 1/\tau \\ 0 & -\alpha & 0 & 0 \\ \beta\tau/\gamma & \beta\tau/\gamma & -\alpha & 0 \\ 0 & 0 & 0 & -\alpha \end{vmatrix} = \alpha^2\left(\alpha^2 - \frac{\beta\tau}{\gamma\tau}\right) = 0$$

Then $\alpha = 0$ or $\alpha = \pm\sqrt{\frac{\beta}{\gamma}}$ and the highest eigenvalue is $\alpha_R = \sqrt{\frac{\beta}{\gamma}}$.

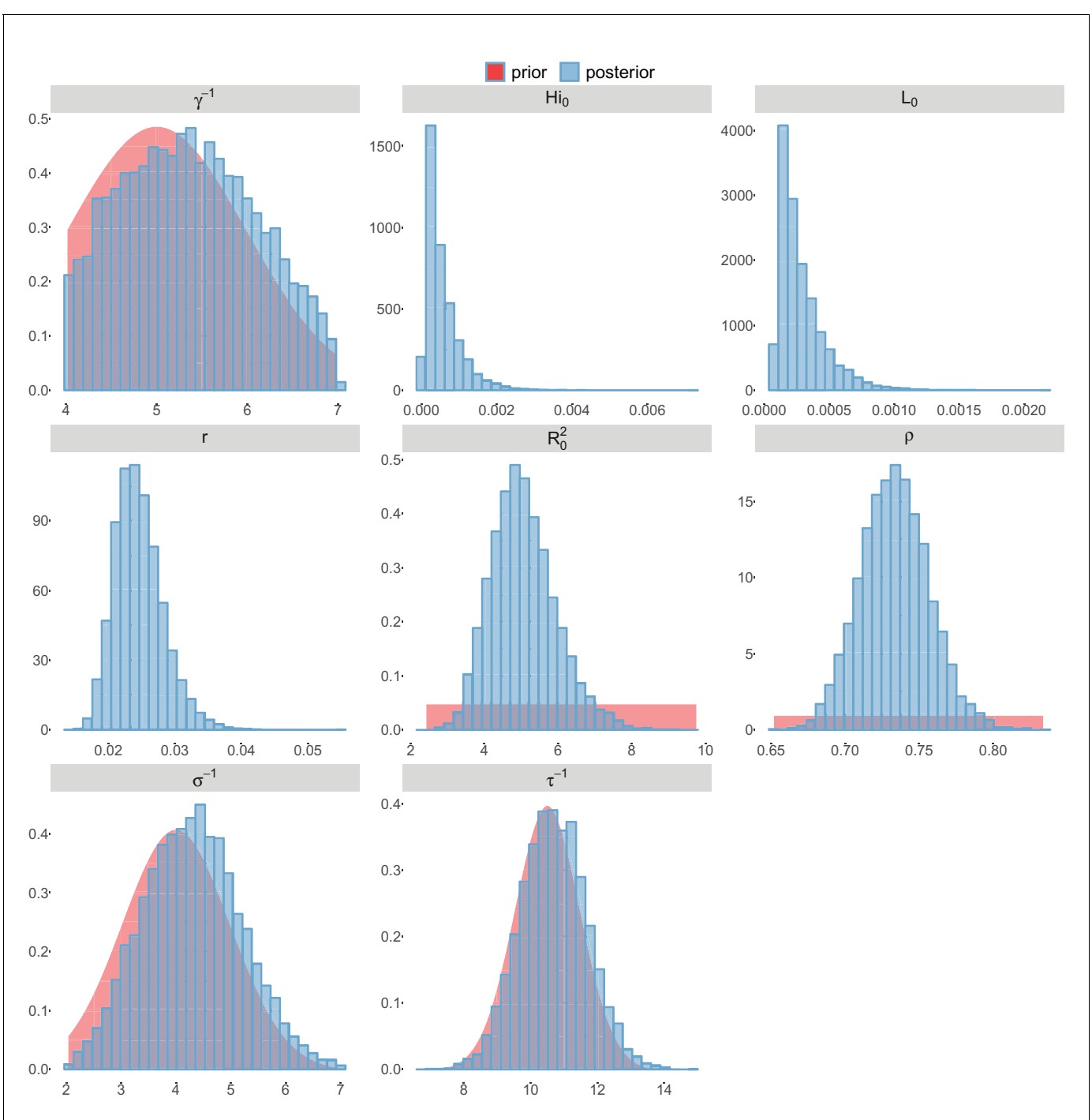

**Figure 10.** Posterior distributions. Laneri model, Yap island.

**Table 5.** Square root of the number of secondary cases after the introduction of a single infected individual in a naive population. Median and 95% credible intervals of 1000 deterministic simulations using parameters from the posterior distribution.

| | Pandey model | Laneri model |
|---|---|---|
| Yap | 3.1 (2.5–4.3) | 2.2 (1.9–2.6) |
| Moorea | 2.6 (2.2–3.3) | 1.8 (1.6–2.0) |
| Tahiti | 2.4 (2.0–3.2) | 1.6 (1.5–1.7) |
| New Caledonia | 2.0 (1.8–2.2) | 1.6 (1.5–1.7) |

Because $\lambda$ and $\kappa$ can be seen as parameters rather than state variables, the interpretation of the spectral radius as the $R_0$ of the model is not straightforward. Therefore, we computed the $R_0$ of the model through simulations, by counting the number of secondary infections after the introduction of a single infected individual in a naive population. Since Laneri model is considered here as a vector model, the number of infected humans after the introduction of a single infected is considered as $R_0^2$. We simulated 1000 deterministic trajectories, using parameter values sampled in the posterior distributions for all parameters except initial conditions. With this method, the confidence intervals for number of infected humans ($R_0^2$) are similar to the ones of $\alpha_R^2$ estimated by the model. As a consequence, $R_0$ was approximated by the spectral radius of the NGM in our results with our stochastic framework (cf. *Table 5*).

As a robustness check, the same method was applied to Pandey model : the confidence intervals for the number of secondary cases in simulations are very similar to the ones of $R_0^2$ (cf. *Table 5*).

## Sensitivity analysis

In order to analyse the influence of parameter values on the model's outputs, a sensitivity analysis was performed, using LHS/PRCC technique (*Blower and Dowlatabadi, 1994*), on Tahiti example. Similar results were obtained for the other settings. Three criteria were retained as outputs for the analysis: the seroprevalence at the last time point, the intensity of the peak of the outbreak and the

**Table 6.** Sensitivity analysis in Pandey model. Tahiti island. 1000 parameter sets were sampled with latin hypercube sampling (LHS), using 'lhs' R package (*Carnell, 2016*). On each parameter set, the model was simulated deterministically in order to explore variability in parameters without interference with variations due to stochasticity. PRCC were computed using the 'sensitivity' R package (*Pujol et al., 2016*).

| Parameters | Distribution | Seroprevalence | Peak intensity | Peak date |
|---|---|---|---|---|
| **Model parameters** | | | | |
| $R_0^2$ | Uniform[0,20] | 0.87 | 0.90 | −0.55 |
| $\beta_V$ | Uniform[0,10] | −0.66 | −0.73 | 0.35 |
| $\gamma^{-1}$ | Uniform[4,7] | −0.25 | 0.10 | 0.20 |
| $\sigma^{-1}$ | Uniform[2,7] | −0.03 | −0.10 | 0.15 |
| $\tau^{-1}$ | Uniform[4,20] | −0.05 | −0.07 | 0.06 |
| $\mu^{-1}$ | Uniform[4,30] | −0.56 | −0.70 | 0.49 |
| **Initial conditions** | | | | |
| $H_I(0)$ | Uniform[2.10⁻⁵,0.011] | 0.05 | −0.02 | 0.02 |
| $V_I(0)$ | Uniform[10⁻⁴,0.034] | 0.11 | −0.00 | −0.26 |
| **Fraction involved and observation model** | | | | |
| $\rho$ | Uniform[0.46,0.54] | 0.47 | 0.15 | −0.03 |
| $r$ | Uniform[0.048,0.072] | −0.04 | 0.03 | 0.05 |

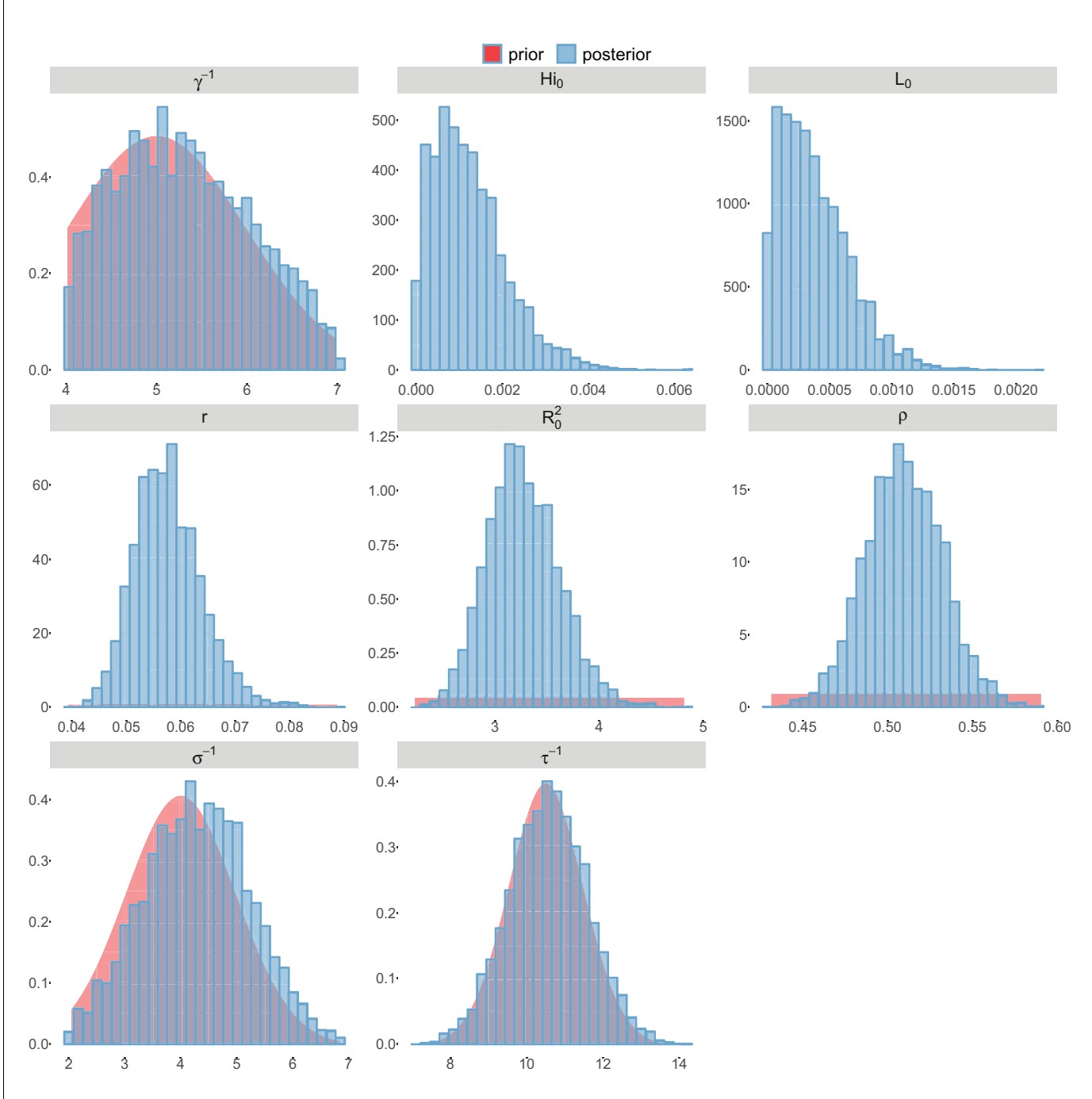

**Figure 11.** Posterior distributions. Laneri model, Moorea island.

date of the peak. We used uniform distributions for all parameters, which are listed in *Tables 6* and *7*. For model parameters, we used the same range as for the prior distribution. For initial conditions, the observation rate $r$ and the fraction involved in the epidemic $\rho$, we used the 95% confidence interval obtained by PMCMC, in order to avoid unrealistic scenarios.

For all criteria, the key parameters in both models are transmission parameters ($R_0$ and $\beta_v$). High values for $R_0$ are positively correlated with a large seroprevalence and a high and early peak. On the contrary, high values for the parameters introducing a delay in the model, the incubation periods in

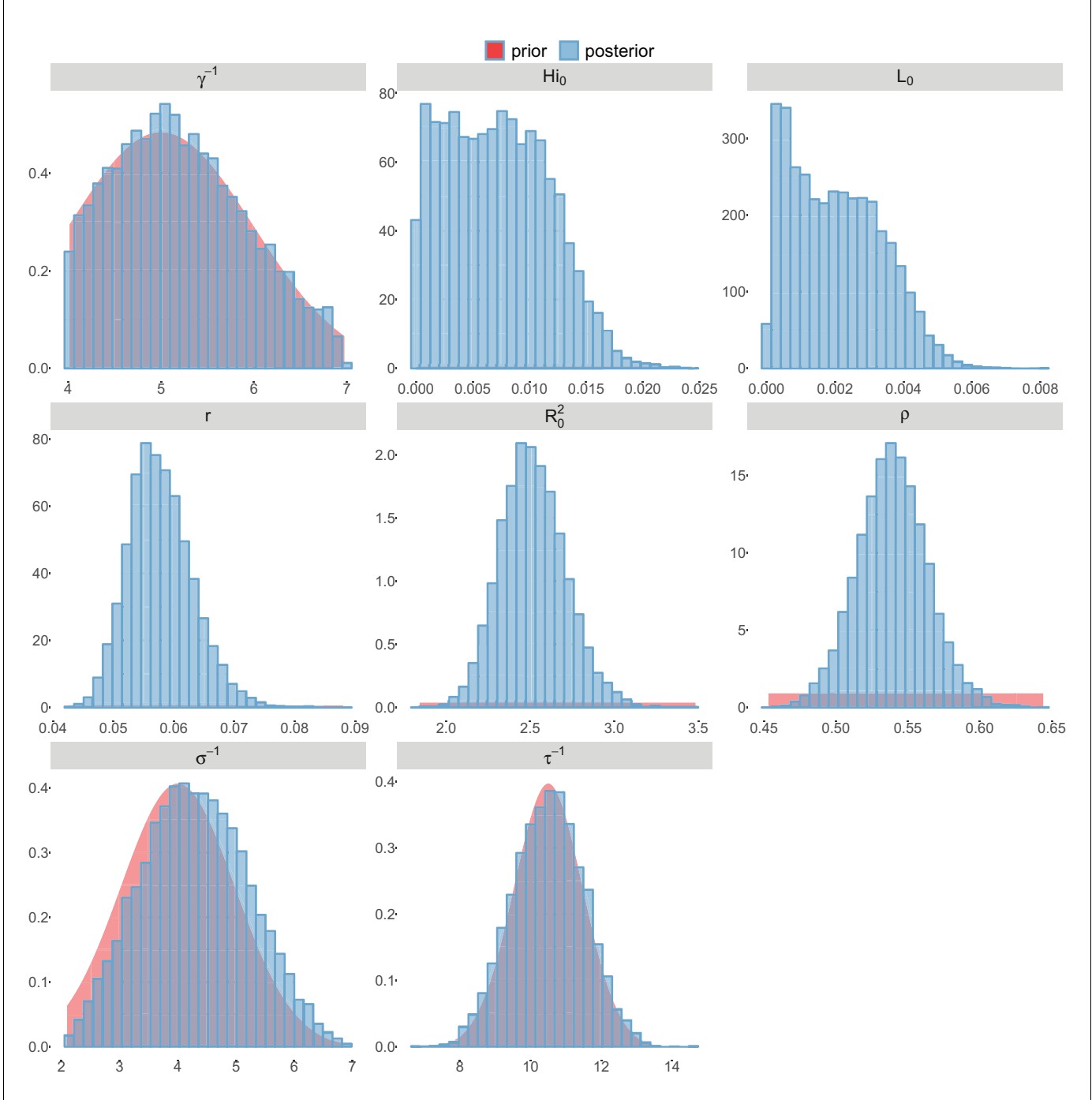

**Figure 12.** Posterior distributions. Laneri model, Tahiti island.

human ($\sigma^{-1}$) and in mosquito ($\tau^{-1}$), are associated with a lower and later peak, and have no significant effect on seroprevalence. Moreover, the simulations are clearly sensitive to the other model parameters, in particular the mosquito lifespan ($\mu^{-1}$) in Pandey model.

Concerning other parameters, the initial conditions have a noticeable effect on the date of the peak only. As expected, the fraction involved in the epidemic ($\rho$) influences the magnitude of the outbreak, by calibrating the proportion of people than can be infected, but it has no significant effect on the timing of the peak.

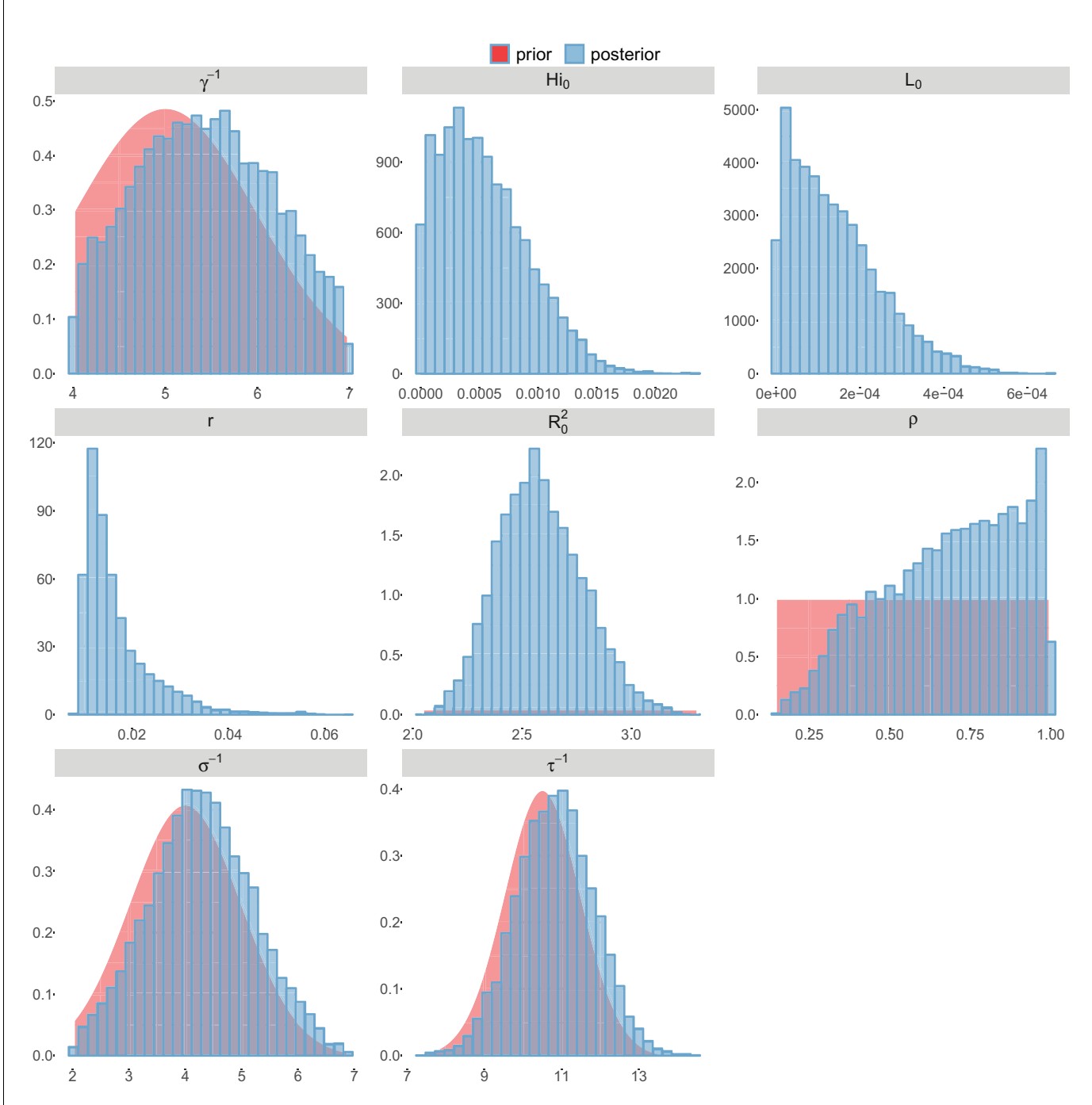

**Figure 13.** Posterior distributions. Laneri model, New Caledonia.

## Complementary results

These complementary results include PMCMC results for both models in the four settings: the epidemic trajectories regarding the human compartments for infected and recovered individuals (*Figures 4,5*), the detailed posterior distributions for all parameters (*Figures 6–13*) and correlation plots for all models (*Figures 14–21*).

**Table 7.** Sensitivity analysis in Laneri model. Tahiti island. 1000 parameter sets were sampled with latin hypercube sampling (LHS), using 'lhs' R package (**Carnell, 2016**). On each parameter set, the model was simulated deterministically in order to explore variability in parameters without interference with variations due to stochasticity. PRCC were computed using the 'sensitivity' R package (**Pujol et al., 2016**).

| Parameters | Distribution | Seroprevalence | Peak intensity | Peak date |
|---|---|---|---|---|
| **Model parameters** | | | | |
| $R_0^2$ | Uniform[0,20] | 0.62 | 0.93 | −0.50 |
| $\gamma^{-1}$ | Uniform[4,7] | 0.01 | 0.62 | 0.15 |
| $\sigma^{-1}$ | Uniform[2,7] | −0.03 | −0.54 | 0.21 |
| $\tau^{-1}$ | Uniform[4,20] | −0.03 | −0.70 | 0.47 |
| **Initial conditions** | | | | |
| $H_I(0)$ | Uniform[$10^{-5}$,0.015] | 0.05 | 0.02 | −0.32 |
| $L(0)$ | Uniform[$2.10^{-5}$,0.004] | 0.05 | 0.00 | −0.16 |
| **Fraction involved and observation model** | | | | |
| $\rho$ | Uniform[0.49,0.59] | 0.80 | 0.34 | 0.02 |
| $r$ | Uniform[0.048,0.068] | −0.01 | 0.01 | −0.02 |

## Correlation between estimated parameters

The inference technique may fail to estimate some parameters due to identifiability issues. In particular, when two parameters are highly correlated to one another, the model manages to estimate the pair of parameters but not each one separately. The analysis of correlation between parameters' posterior distributions can reveal such cases. The following graphics display for each model the correlation coefficients between all pairs of parameters across the MCMC chain. For example, in models for New Caledonia, the observation rate and the fraction of the population involved in the epidemic

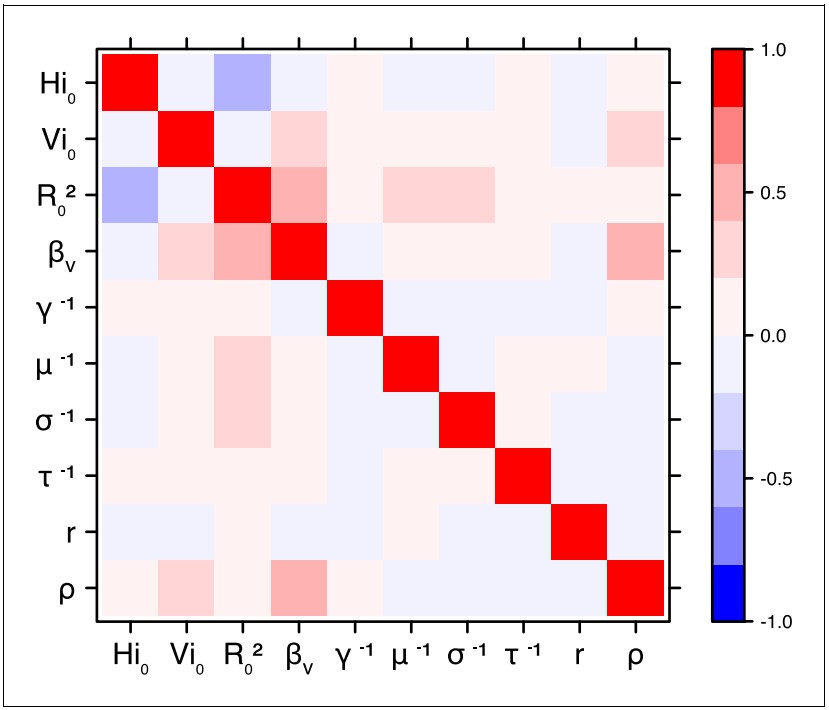

**Figure 14.** Correlation plot of MCMC output. Pandey model, Yap island.

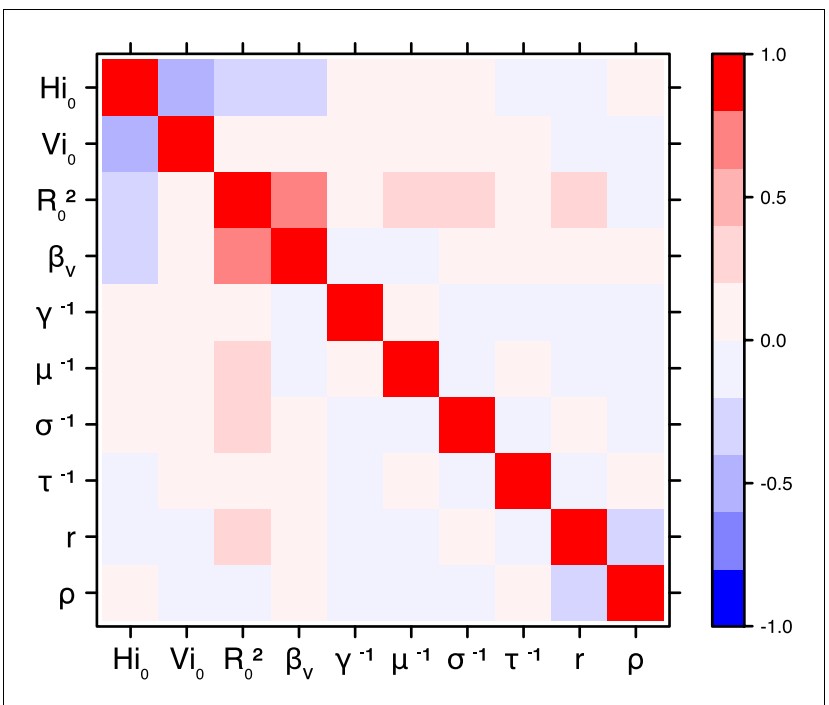

**Figure 15.** Correlation plot of MCMC output.  Pandey model, Moorea island.

are strongly negatively correlated (*Figures 17*,*21*): the inference technique does not manage to estimate properly these two parameters, due to the lack of information on seroprevalence.

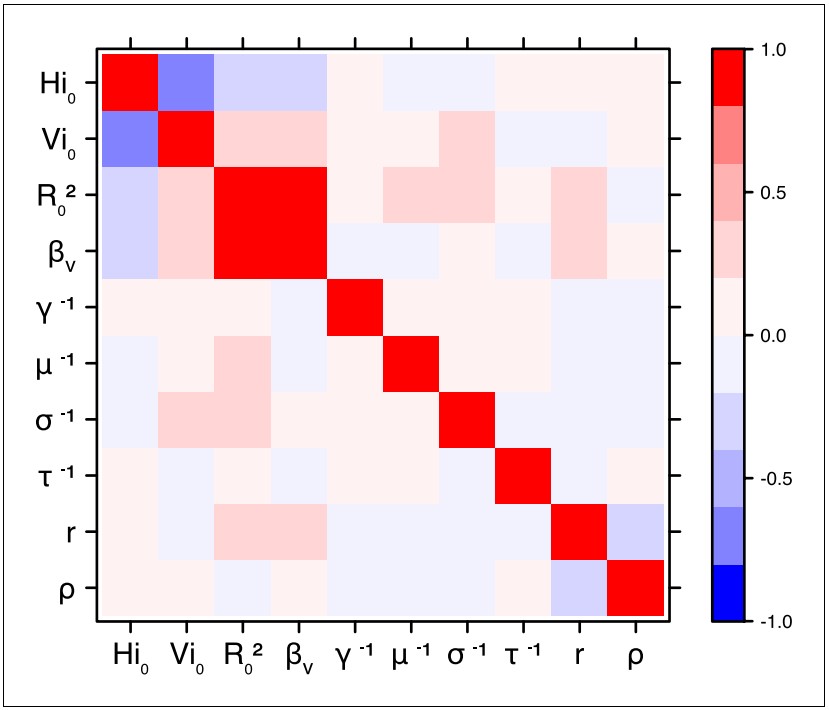

**Figure 16.** Correlation plot of MCMC output.  Pandey model, Tahiti island.

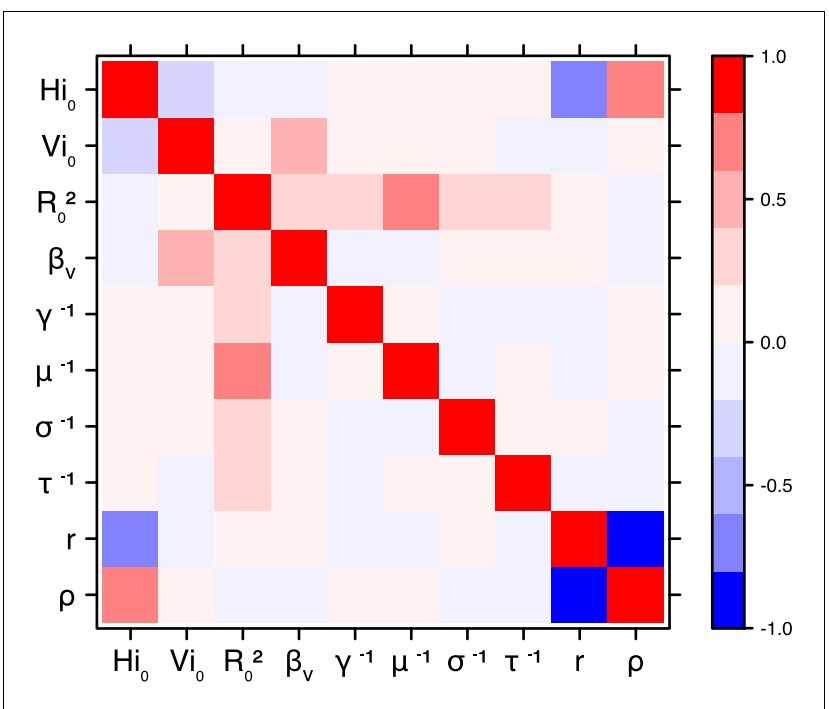

**Figure 17.** Correlation plot of MCMC output. Pandey model, New Caledonia.

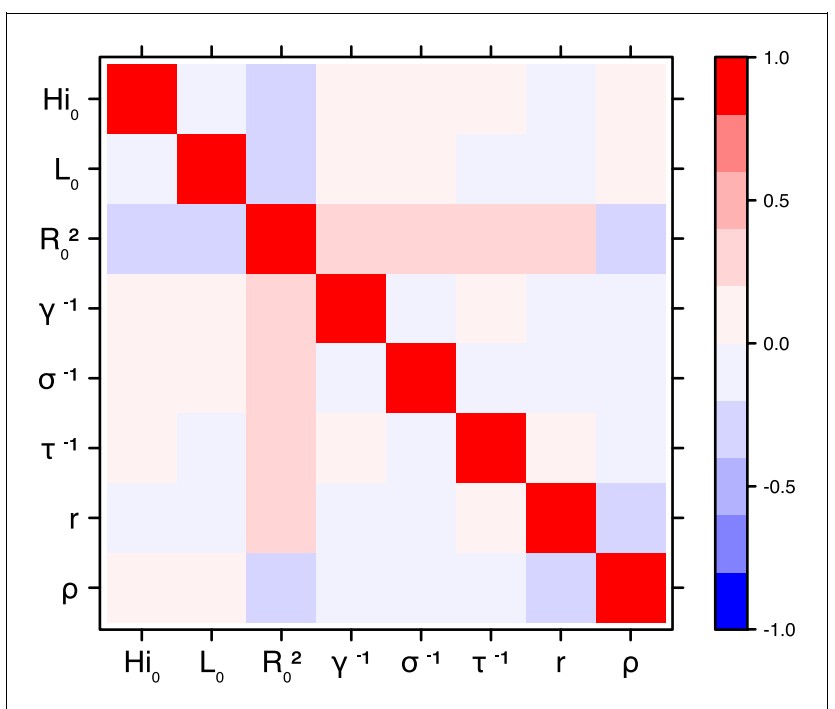

**Figure 18.** Correlation plot of MCMC output. Laneri model, Yap island.

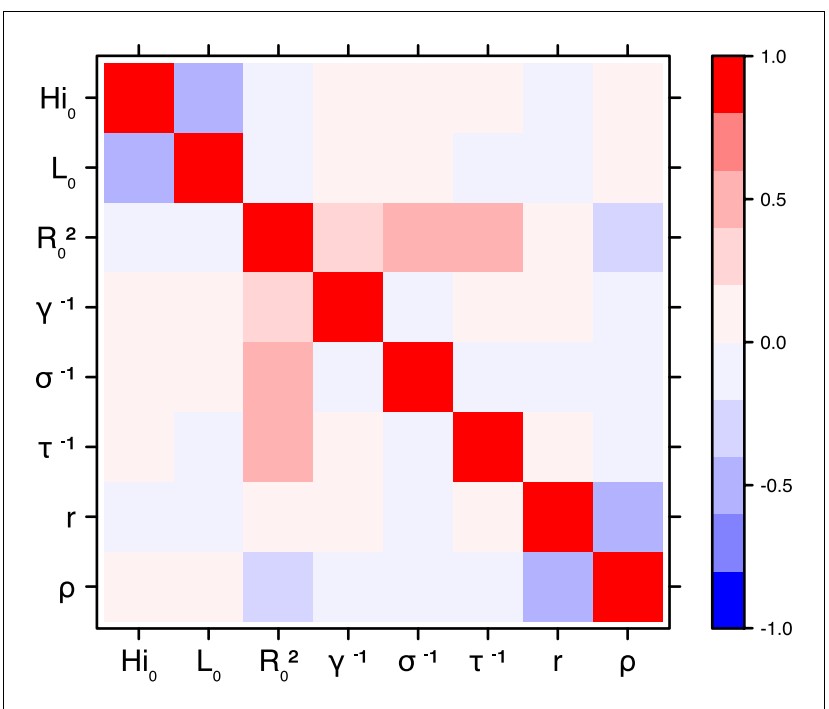

**Figure 19.** Correlation plot of MCMC output. Laneri model, Moorea island.

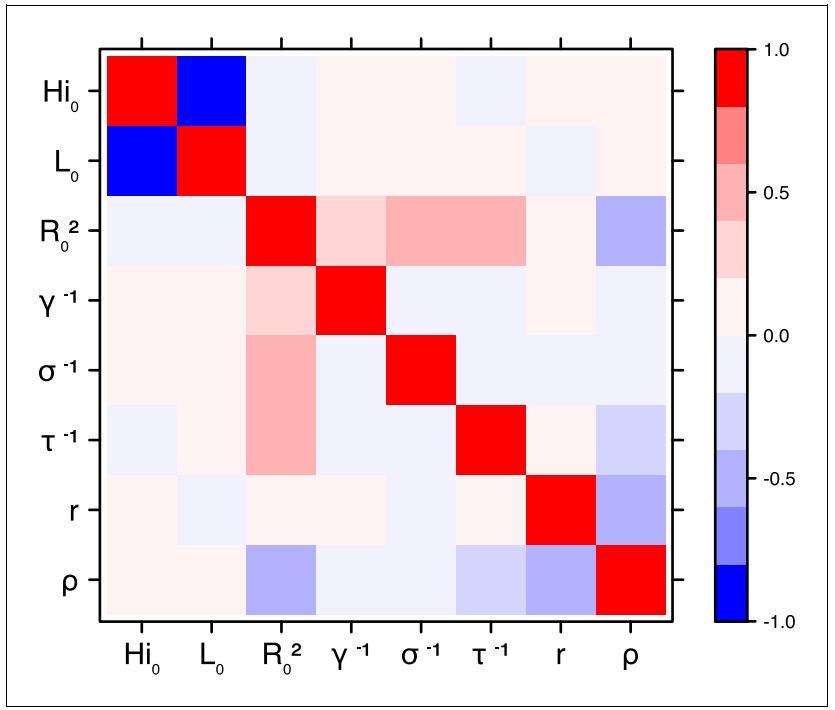

**Figure 20.** Correlation plot of MCMC output. Laneri model, Tahiti island.

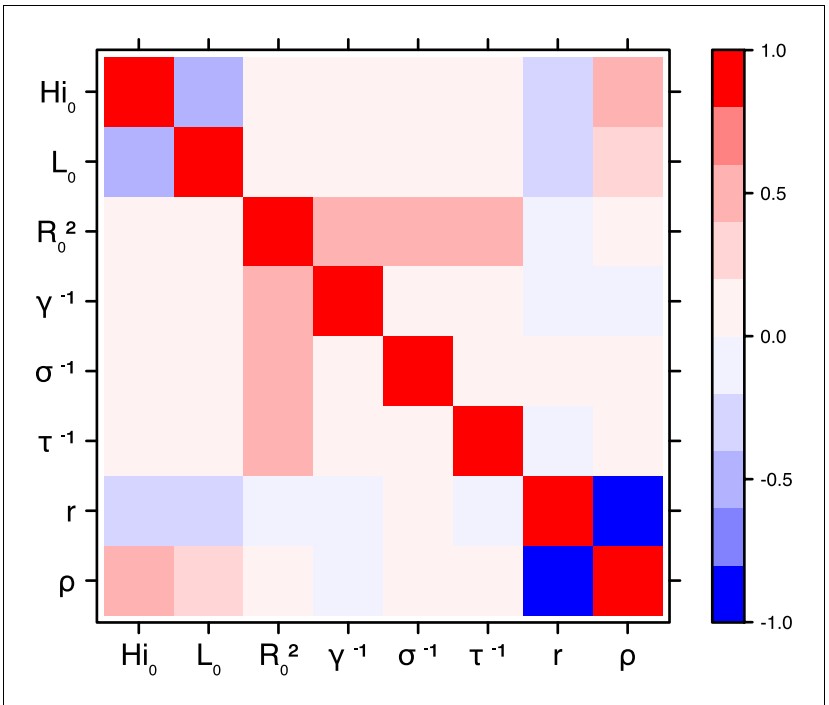

**Figure 21.** Correlation plot of MCMC output. Laneri model, New Caledonia.

## Code and source data files

The estimation tools are provided by the open source software SSM (*Dureau et al., 2013*) (State Space Models, RRID:SCR_014647), available at https://github.com/JDureau/ssm and https://github.com/sballesteros/ssm-predict. The codes for the implementation of each model are provided as supplementary file.

## Acknowledgements

CC, DGS and BC are partially supported by the "Pépiniere interdisciplinaire Eco-Evo-Devo" from the Centre National de la Recherche Scientifique (CNRS). The research leading to these results has also received funding from the European Commission Seventh Framework Program [FP7/2007–2013] for the DENFREE project under Grant Agreement 282 378. The funders played no role in the study design, data collection, analysis, or preparation of the manuscript.

## Additional information

### Funding

| Funder | Grant reference number | Author |
| --- | --- | --- |
| Centre National de la Recherche Scientifique | Pepiniere interdisciplinaire Eco-Evo-Devo | Clara Champagne<br>David Georges Salthouse<br>Bernard Cazelles |
| European Commission | Seventh Framework Program [FP7 2007-2013] for the DENFREE project under Grant Agreement 282 348 | Clara Champagne<br>David Georges Salthouse<br>Richard Paul<br>Van-Mai Cao-Lormeau<br>Bernard Cazelles |

The funders had no role in study design, data collection and interpretation, or the decision to submit the work for publication.

## Author contributions

CC, BC, Contributed to the study design, numerical part of the study, interpretation of the results and writing of the manuscript; DGS, Contributed to the numerical part of the study, interpretation of the results and writing of the manuscript; RP, V-MC-L, BR, Contributed to the interpretation of the results and writing of the manuscript

## Author ORCIDs

Clara Champagne, http://orcid.org/0000-0002-0369-6758
Bernard Cazelles, http://orcid.org/0000-0002-7972-361X

## Additional files

### Supplementary files

• Supplementary file 1. Codes for the implementation of each model.

• Supplementary file 2. Data file, Yap island. Number of cases per week and number of immune individuals at the end of the epidemic (*Duffy et al., 2009*).

• Supplementary file 3. Data file, Moorea island. Number of cases per week and number of immune individuals at the end of the epidemic (*Mallet et al., 2015*; *Aubry et al., 2015b*).

• Supplementary file 4. Data file, Tahiti island. Number of cases per week and number of immune individuals at the end of the epidemic (*Mallet et al., 2015*; *Aubry et al., 2015b*).

• Supplementary file 5. Data file, New Caledonia. Number of cases per week (*DASS, 2014*).

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
