## [Decision Letter]

Thank you for submitting your article "Structure in the variability of the basic reproductive number (*R*_0_) for Zika epidemics in the Pacific islands" for consideration by *eLife*. Your article has been reviewed by two peer reviewers, one of whom is a member of our Board of Reviewing Editors, and the evaluation has been overseen by Prabhat Jha as the Senior Editor. The reviewers have opted to remain anonymous.

The reviewers have discussed the reviews with one another and the Reviewing Editor has drafted this decision to help you prepare a revised submission.

On the whole, the reviewers and the Reviewing Editor appreciated the work. Although there have been a number of previous attempts to estimate *R*_0_ for Zika using Pacific Island data, they see the importance in the use of a stochastic model in order to obtain more precise values for both point estimates and uncertainty bounds, as well as the efforts to capture some degree of the structural uncertainty in model design. Making the source data available was particularly appreciated as this enhances the transparency and reproducibility of the analysis.

The reviewers and Reviewing Editor had only two requirements, although both of them are important:

1) Despite a better effort than many to explain the methods used, the details were still unclear to the reviewers. Hence we would like to see the full model code used for the analysis made available alongside the source data.

2) There was some concern over the estimation and interpretation of the parameter *ρ*, which appears to be a key parameter in the model but is poorly defined. This reduces the population actually involved in the epidemic dynamics as a result of "spatial heterogeneity, immuno-resistance or cross-immunity". It ranges between 34% to 74% across models and islands, and is especially variable in New Caledonia between the two models (34% – 70%). It seems particularly strange that *ρ* is actually smaller than final seroprevalence rate in the outbreak (73% in Yap and 94% in French Polynesia), when most of the population was naïve prior to the outbreak. How do people seroconvert if they are not involved in the epidemic? And if part of the population was actually protected from infection, is there any evidence to support the mechanism (eg. genetic links to Zika susceptibility, or spatial data showing that part of the islands were unaffected)?

[Editors' note: further revisions were requested prior to acceptance, as described below.]

Thank you for resubmitting your work entitled "Structure in the variability of the basic reproductive number (*R*_0_) for Zika epidemics in the Pacific islands" for further consideration at *eLife*. Your revised article has been favorably evaluated by Prabhat Jha as the Senior editor and the previous reviewers.

Firstly, we would like to apologise for the time it has taken to respond to your resubmission. We wanted to consult further about the issue of model code availability stated below, but unfortunately this process took far longer than anticipated.

Overall we are happy with the revised article and are almost ready to proceed to acceptance. The one outstanding issue is the availability of the model code. We do understand that you have provided a link to the code to the algorithm used for parameter inference. However, the editors and reviewer were unable to understand how to reproduce your numerical findings using the code in the repository plus the data included in the Appendix. As far as we can tell, all you have provided is a link to code written by other developers that was used in your model, not the actual compartmental model itself.

We would like you to provide either (i) an explanation that we are wrong and that your results (including all the figures behind all the tables and graphs) can be fully regenerated using this code, or (ii) the actual code itself. Once you are able to provide this, we should be able to proceed very rapidly to a final decision

---

## [Author Response]

*On the whole, the reviewers and the Reviewing Editor appreciated the work. Although there have been a number of previous attempts to estimate R_0_ for Zika using Pacific Island data, they see the importance in the use of a stochastic model in order to obtain more precise values for both point estimates and uncertainty bounds, as well as the efforts to capture some degree of the structural uncertainty in model design. Making the source data available was particularly appreciated as this enhances the transparency and reproducibility of the analysis.*

*The reviewers and Reviewing Editor had only two requirements, although both of them are important:*

*1) Despite a better effort than many to explain the methods used, the details were still unclear to the reviewers. Hence we would like to see the full model code used for the analysis made available alongside the source data.*

We added the links to the open source software SSM, and the JSON codes for the implementation of each model.

*2) There was some concern over the estimation and interpretation of the parameter ρ, which appears to be a key parameter in the model but is poorly defined. This reduces the population actually involved in the epidemic dynamics as a result of "spatial heterogeneity, immuno-resistance or cross-immunity". It ranges between 34% to 74% across models and islands, and is especially variable in New Caledonia between the two models (34% – 70%). It seems particularly strange that ρ is actually smaller than final seroprevalence rate in the outbreak (73% in Yap and 94% in French Polynesia), when most of the population was naïve prior to the outbreak. How do people seroconvert if they are not involved in the epidemic? And if part of the population was actually protected from infection, is there any evidence to support the mechanism (eg. genetic links to Zika susceptibility, or spatial data showing that part of the islands were unaffected)?*

We provided an extended description of the parameter *ρ*, both in the Methods and Results sections:

“The fraction *ρ* of the population involved in the epidemic is well estimated when the seroprevalence is known (in Yap and French Polynesia). In these cases, the proportion of the population involved is slightly greater than the seroprevalence rate, indicating a very high infection rate among involved individuals. In New Caledonia, as no information on seroprevalence was available, the fraction of population involved displays very large confidence intervals (cf. Table 1 and Table 2), indicating that the model did not manage to identify properly this parameter with the available data. In this case, this parameter is highly correlated to the observation rate *r* (Figure 17 and Figure 21): *r* and *ρ* seem unidentifiable without precise information on seroprevalence.”

“In both models, following the dominant paradigm that diseases transmitted by *Aedes* mosquitoes are highly clustered, we restricted the total population H measured by census to a fraction *N = ρ.H*, in which the parameter *ρ* is estimated. This formulation makes the hypothesis that a fraction of the total population is not at risk from the epidemic, because of individual factors or because the individuals remain in areas where the virus is not present. Moreover, as the vector’s flying range is small, the clustering of ZIKV infection may be reinforced. This may result in escapees from infection within the population, even at a single island scale. The available data does not allow further exploration of the mechanisms underlying these phenomena, which seem fundamental to understand ZIKV propagation. At the very least, the restriction to a fraction *ρ* enables the model to reproduce the observed seroprevalence rates, and to provide coherent results with respect to both data sources (seroprevalence and surveillance data).”

In all our estimations, the fraction *ρ* is greater than seroprevalence. The final seroprevalence rate in French Polynesia was 49% (Aubry et al. 2015) and 73% in Yap (Duffy et al. 2009) and *ρ* was estimated respectively to 50% and 74%. Because no seroprevalence information was available for New Caledonia, the estimated *ρ* varied between models.

Posterior estimates for the parameters are calculated on the MCMC chain, and estimated seroprevalence is calculated on simulations of the model with parameters sampled in the MCMC chain. Our initial code used different lengths of the MCMC chain, generating some minor problems in the estimation of the medians. This did not impact the results, except for *ρ* in New Caledonia, because, due to the lack of seroprevalence data, *ρ* is poorly identifiable, and its posterior distribution is very skewed. We corrected this imprecision by using the same chain length: now all our estimates use a chain with 50,000 iterations.

In all cases, people do not seroconvert if they are not involved in the epidemic. The precise mechanisms by which a part of the population is protected from infection are unknown, even though there is some evidence of a spatial clustering of cases, as it was observed for dengue in other contexts (see for example Telle et al. PLoS One 2015). For instance, in French Polynesia, previous dengue infection rates display large spatial variations even within islands (cf. Daudens et al. “Épidémiologie de la dengue et stratégies de lutte en Polynésie française, 2006-2008”, Bulletin épidémique hebdomadaire, InVS, 2009, December 22th).

*[Editors' note: further revisions were requested prior to acceptance, as described below.]*

*Overall we are happy with the revised article and are almost ready to proceed to acceptance. The one outstanding issue is the availability of the model code. We do understand that you have provided a link to the code to the algorithm used for parameter inference. However, the editors and reviewer were unable to understand how to reproduce your numerical findings using the code in the repository plus the data included in the appendix. As far as we can tell, all you have provided is a link to code written by other developers that was used in your model, not the actual compartmental model itself.*

*We would like you to provide either (i) an explanation that we are wrong and that your results (including all the figures behind all the tables and graphs) can be fully regenerated using this code, or (ii) the actual code itself. Once you are able to provide this, we should be able to proceed very rapidly to a final decision*

Firstly, we want to thank the Editor and the reviewers for their remarks and their effort to reproduce our calculation. We understand that the use of our codes is not straightforward and therefore, we now provide more detailed files and explanations.

The code in the online repository is the software SSM: it is the library to be installed in order to perform the estimations using the pmcmc and kmcmc algorithms. In our previous resubmission, we also provided the.json files describing the models (as a zip “Source Code” file). We extended this zip folder, with some initialisation and result files, and we added a READ ME document describing the procedure to be followed in order to use the software and our codes. We also added an R code for plotting and an example of our estimations (for Pandey model in Tahiti), so that it can be plotted directly.

Finally, we made the reference to this folder more explicit in the manuscript (it is now referenced as “[Supplementary-material SD1-data]”).

We reproduce here the text of the READ ME file:

1) Install SSM software on your machine as indicated in https://github.com/JDureau/ssm; the complementary module “SSM predict” also needs to be installed (https://github.com/sballesteros/ssm-predict)

2) In order to run one model, four elements are needed:

a) Data files in csv format

b) Priors distributions in json format

c) An initialisation file in json format

d)The json file containing the model, called “ssm.json”

These 4 elements are provided for each model, in a file with the corresponding names. The initialisation file provided here is the result of a previous simplex run (called theta_simplex.json) but one can start with other initial values (also put in a file named theta_simplex.json for using with the given code).

3) Execute the R code “pmcmc_kmcmc_simulations.R”.

This files enables to compile the model, run KMCMC and PMCMC algorithm and do a simulation of the model (using our estimated values).

The PMCMC part is to be used when convergence is reached with KMCMC algorithm, which might sometimes require more than 5 KMCMC steps.

The PMCMC part is computationally intensive and may require a cluster (when the number of particles is 10,000).

We provided our estimates for Pandey model in Tahiti as an example, in the “pmcmc-result” file: they comprise the last PMCMC chain (trace_0.csv), the filter trajectory (X_0.csv), the last set of parameters of the chain (mle.json) and the simulation of the model (X_simul.csv)

4) All the calculation is done, and the outputs can be plotted using for example the R code “plotting”.